# Sparse-to-Sparse Training of Diffusion Models

**Inês Cardoso Oliveira**                                    *i.oliveira@uni.lu*
*University of Luxembourg*

**Decebal Constantin Mocanu**                          *decebal.mocanu@uni.lu*
*University of Luxembourg*

**Luis A. Leiva**                                            *luis.leiva@uni.lu*
*University of Luxembourg*

**Reviewed on OpenReview:** *https://openreview.net/forum?id=iRupdoPLJa*

## Abstract

Diffusion models (DMs) are a powerful type of generative models that have achieved state-of-the-art results in various image synthesis tasks and have shown potential in other domains, such as natural language processing and temporal data modeling. Despite their stable training dynamics and ability to produce diverse high-quality samples, DMs are notorious for requiring significant computational resources, both in the training and inference stages. Previous work has focused mostly on increasing the efficiency of model inference. This paper introduces, for the first time, the paradigm of sparse-to-sparse training to DMs, with the aim of improving both training and inference efficiency. We focus on unconditional generation and train sparse DMs from scratch (Latent Diffusion and ChiroDiff) on six datasets using three different methods (Static-DM, RigL-DM, and MagRan-DM) to study the effect of sparsity in model performance. Our experiments show that sparse DMs are able to match and often outperform their Dense counterparts, while substantially reducing the number of trainable parameters and FLOPs. We also identify safe and effective values to perform sparse-to-sparse training of DMs.

## 1 Introduction

Diffusion models (DMs) are a class of deep generative models that exhibit extraordinary performance to produce diverse and high-quality data. DMs currently dominate the generative field in computer vision, having been applied to a wide range of tasks such as (un)conditional image generation (Ho et al., 2020b; Rombach et al., 2021; Nichol & Dhariwal, 2021; Dhariwal & Nichol, 2021; Nichol et al., 2022; Blattmann et al., 2022; Das et al., 2023), image super-resolution (Saharia et al., 2021; Chung et al., 2022), and image inpainting (Nichol et al., 2022; Chung et al., 2022; Saharia et al., 2022), among others. DMs have also shown incredible potential in other domains, including speech generation (Liu et al., 2023a), text generation (Li et al., 2022; Gong et al., 2023), and time-series prediction and imputation (Rasul et al., 2021; Tashiro et al., 2021).

Despite these advantages, DMs are notorious for their slow training, demanding significant computational resources and resulting in a considerable carbon footprint (Strubell et al., 2020). Due to the extensive number of diffusion timesteps required to produce a single sample (e.g., Rombach et al. (2021) mentioned up to 500 steps), DMs also suffer from slow sampling speed (Song et al., 2021). Even though progress has been made in improving inference speed, DMs are still considerably slower than other generative approaches such as GANs and VAEs (Rombach et al., 2021). This inefficiency impacts not only end users, but also the research community, by hindering further developments due to the lengthy process of model training and evaluation.

Reducing the computational costs and memory requirements of DMs is a critical challenge for the broad implementation and adoption of these models, and an active field of research. Much of the existent literature

has addressed this challenge through improvements to the inference stage (Song et al., 2021; Nichol & Dhariwal, 2021; Fang et al., 2023; Shang et al., 2023; Li et al., 2023; Salimans & Ho, 2022; Meng et al., 2023). Efforts have also been made in the direction of training efficiency, exploring different architectures and training strategies (Wang et al., 2023; Ding et al., 2023; Rombach et al., 2021; Phung et al., 2022), but training DMs is still an extensive and costly process.

In the last few years, sparse-to-sparse training has emerged as a promising approach to significantly reduce the computational cost of deep learning models, by training sparse networks from scratch (Mocanu et al., 2018; Bellec et al., 2018; Dettmers & Zettlemoyer, 2019; Evci et al., 2020; Zhang et al., 2024b). Interestingly, sparse neural networks have been shown to match, or even outperform, their Dense counterparts in classification tasks (Mocanu et al., 2018; Liu et al., 2021a), generative modeling using GANs (Liu et al., 2023b), and Reinforcement Learning (Sokar et al., 2022), all while requiring less memory and reducing the number of floating-point operations (FLOPs). We should note that, currently, most sparse neural networks require roughly the same amount of time to train as their dense counterparts, since today's hardware is optimized for dense matrix operations. However, growing interest in sparse models is reshaping the landscape; see Appendix A for a discussion in this regard.

We propose to lower the computational cost of DMs by incorporating, for the first time, the paradigm of sparse-to-sparse training for unconditional generation. As such, we introduce three different methods, Static-DM (static strategy), RigL-DM, and MagRan-DM (both dynamic strategies), that can be easily integrated with existing DMs. Since our goal is to study the effect of these techniques on the performance of DMs, we experiment using two state-of-the-art DMs in two domains: Latent Diffusion (Rombach et al., 2021) for image generation (continuous, pixel-level data) and ChiroDiff (Das et al., 2023) for sketch generation (discrete, spatiotemporal sequence data). In sum, we make the following contributions:

- We introduce sparse-to-sparse training to unconditional DMs, with both static and dynamic strategies. We consider various sparsity levels, two state-of-the-art models (Latent Diffusion and ChiroDiff), and six datasets in total. We also perform experiments using conditional DMs.

- Our experiments show great promise of sparse-to-sparse training for DMs, as we were able to train a sparse DM for each model/dataset case with comparable performance to their respective Dense counterpart, while significantly reducing the parameters count and FLOPs. In most cases, at least one sparse DM outperformed its Dense version.

- We identify safe and effective values to perform sparse-to-sparse training of DMs. Higher performance is achieved using dynamic sparse training with 25–50% sparsity levels. For models with higher sparsity ratio, a conservative prune and regrowth ratio of 0.05 provides better results.

## 2 Background and Related Work

### 2.1 Diffusion Models

DMs (Sohl-Dickstein et al., 2015; Ho et al., 2020a; Song et al., 2020) are probabilistic models designed to learn a data distribution $q(x)$ through two processes: a forward noising process and a reverse denoising process. The forward process is defined as a Markov Chain of length $T$ in which Gaussian noise is added at each timestep $t$, producing a sequence of increasingly noisier samples:

$$q(x_t|x_{t-1}) = \mathcal{N}(x_t; \sqrt{1 - \beta_t}x_{t-1}, \beta_t\mathbf{I}) \tag{1}$$

$$q(x_{1:T}|x_0) = \prod_{t=1}^{T} q(x_t|x_{t-1}) \tag{2}$$

where $x_0$ is the original data point, $x_t$ is the data point at timestep $t$, and $\beta_t$ is the pre-defined amount of noise added at timestep $t$.

The reverse denoising process $q(x_{t-1}|x_t)$, attempts to recover the original data, but it is intractable as it depends on the entire data distribution $q(x)$. As such, we need to parameterize a neural network $p_\theta$

to approximate it. This network $p_\theta$ can be optimized by training with the simplified objective, $\mathcal{L} = \mathbb{E}_{t\sim[1,T],x_0,\epsilon\sim\mathcal{N}(0,1)}||\epsilon - \epsilon_\theta(x_t,t)||^2$, where $x_T$ is a noisy version of input $x$ at the final timestep $T$, and $\epsilon_\theta$ the prediction of the neural network $p_\theta$.

## 2.2 Efficiency in Diffusion Models

Increasing the efficiency of DMs has been primarily addressed through accelerating the sampling process, by reducing the number of diffusion steps through faster sampling (Song et al., 2021; Karras et al., 2022) and model distillation (Salimans & Ho, 2022; Meng et al., 2023; Yin et al., 2024). As for training acceleration, some works have proposed shifting the diffusion process to the latent space (Rombach et al., 2021; Vahdat et al., 2021). Interestingly, Phung et al. (2022) used discrete wavelet transforms to decompose images into sub-bands, employing these sub-bands to perform the diffusion more efficiently.

Previous studies have also presented refinements to the training process of DMs. For example, Wang et al. (2023) introduced a plug-and-play training strategy that utilizes patches instead of the full images, to improve training speed. Hang et al. (2024) proposed treating DMs as a multitask learning problem and introduced a weighting strategy to balance the different timesteps, achieving a significant improvement in training convergence speed.

From the perspective of network compression, prior works have explored techniques such as structural pruning (Fang et al., 2023), post-training quantization (Shang et al., 2023; Li et al., 2023), knowledge distillation (Yang et al., 2023), and the lottery ticket hypothesis (Frankle & Carbin, 2019; Jiang et al., 2023). Very recently, Wang et al. (2024) proposed the incorporation of sparse masks into pre-trained DMs before fine-tuning, and achieved a 50% reduction in multiply-accumulate operations (MACs) with only a slight average decrease of image quality (as measured by the FID score). Although these techniques work in increasing efficiency, they still require pre-training of full DMs. Our work proposes training sparse DMs from scratch, which has the potential to both accelerate training and inference, and reduce the memory footprint.

## 2.3 Sparse-to-Sparse Training

Nowadays most computational models are what is referred to as *Dense* networks, comprising a stack of layers containing multiple neurons, each connected to all neurons in the following layer. Sparse-to-sparse training techniques aim to train sparse neural networks from scratch, thus reducing the number of parameters and computations. If we define the connectivity graph of a Dense neural network as $G(\mathcal{V},\mathcal{E})$, where $\mathcal{V}$ represents the set of neurons (vertices), and $\mathcal{E}$ the set of connections between them (edges), a sparse version of that neural network would be defined as $G(\mathcal{V}',\mathcal{E}')$, with $\mathcal{V}'$ and $\mathcal{E}'$ being a subset of the neurons and connections of the Dense network. Sparse networks can be obtained using structured methods, where $\mathcal{V} \neq \mathcal{V}'$, and unstructured methods, where $\mathcal{V} = \mathcal{V}'$. Overall, sparse-to-sparse training techniques can be divided into static sparse training (SST) and dynamic sparse training (DST).

**Static Sparse Training.** In SST methods, the connectivity pattern between neurons is set at initialization, and remains fixed during training. This concept was first introduced by Mocanu et al. (2016), who proposed a non-uniform scale-free topology for Restricted Boltzmann machines, with the sparse models achieving better results than their Dense counterparts. Later, Liu et al. (2022) investigated the efficacy of random pruning at initialization, and found that, using appropriate layer-wise sparsity ratios, a randomly pruned subnetwork of WideResNet-50 can outperform a dense WideResNet-50 on ImageNet. Many other criteria have been proposed to set layer-wise sparsity ratios before training, by trying to identify important connections using information such as connection sensitivity, as in SNIP (Lee et al., 2019), gradient flow (Wang et al., 2020), as in GraSP. Very recently, two new initialization criteria have been proposed that utilize concepts from network science theory: Bipartite Scale-Free and Bipartite Small-World (Zhang et al., 2024a;b).

**Dynamic Sparse Training.** In DST methods, the network is initialized with a connectivity pattern and dynamically explores different connections throughout training (Mocanu et al., 2018; Bellec et al., 2018). This was first proposed by Mocanu et al. (2018) through Sparse Evolutionary Training (SET), an algorithm that adjusts the connections using a prune-and-grow scheme every $N$ training steps. In SET, weights are

dropped based on their magnitude (ensuring an equal amount of positive and negative weights) and regrown randomly. RigL (Evci et al., 2020) proposes an alternative method that prunes the weights based on the absolute magnitude, and regrows them based on the gradients by calculating the dense gradients only at the update step. Although further pruning methods have been proposed (Lee et al., 2019; Yuan et al., 2021), a study by Nowak et al. (2023) found only minor differences between the tested criteria. The contrast was higher in lower density patterns, with magnitude pruning giving the best performance. Other growing criteria have been proposed based on randomness (Mostafa & Wang, 2019) and momentum (Dettmers & Zettlemoyer, 2019).

Recently, Zhang et al. (2024b) proposed Epitopological Sparse Meta-deep Learning (ESML), a brain-inspired, gradient-free method, which aims to shift the focus from the weights to the network topology, and uses concepts from network theory. By leveraging ESML, the authors train a sparse network that using just 1% of the connections, is able to surpass dense networks, as well as other DST methods, in several image classification tasks.

DST has also been applied to the field of generative modelling: Liu et al. (2023b) proposed STU-GAN, comprised of a generator with high sparsity and a denser discriminator. STU-GAN was able to outperform a dense BigGAN on CIFAR-10 with a 80% sparse generator and 70% sparse discriminator.

## 3 Methodology

Our study aims to understand the effect of sparse-to-sparse training techniques on DMs. We focus on unstructured sparsity due to its ability to maintain high performance even at very high levels of sparsity (Evci et al., 2020). Thus, our experiments cannot rely on current hardware to accelerate sparse computations; for example, NVIDIA A100 and Ampere cards only support 2:4 structured sparsity, which requires to enforce a fixed sparsity level of 50%. In the following sections, we present three methods of introducing sparsity in DMs: one SST technique, Static-DM, and two DST techniques, MagRan-DM and RigL-DM.

### 3.1 Static Sparse Training: Static-DM

Static-DM is a sparse DM trained from scratch, with fixed connectivity between neurons. The pseudocode for Static-DM is shown in Algorithm 1. The training process closely resembles that of a dense DM, with the addition of a sparse initialization step. In this step, the graph underlying the neural network is sparsified by setting a fraction of the neuron connections to zero.

---

**Algorithm 1** Static-DM

1: **Input:** Dataset $\mathcal{D}$, Network $f_\theta$, Number of Epochs $N$, Diffusion steps $T_d$, Sparsity ratio $S$
2: $\theta \leftarrow$ sparse initialization using $S$
3: **for** $i = 1$ to $N$ **do**
4: $\quad x_0 \sim \mathcal{D}$
5: $\quad t \sim \mathcal{U}(\{1, 2, \ldots, T_d\})$
6: $\quad \epsilon \sim \mathcal{N}(\mathbf{0}, \mathbf{I})$
7: $\quad \theta_i = \text{AdamW}(\nabla_\theta, \mathcal{L}_{\text{DIF}}(f_\theta(x_0, t), \epsilon))$
8: **end for**

---

Following the findings of Liu et al. (2022), we randomly prune the connections at initialization using the Erdõs–Rényi (ER) (Mocanu et al., 2018) strategy to allocate the non-zero weights to non-convolutional layers. With this strategy, larger layers get assigned higher sparsity than smaller layers. The sparsity of each layer scales with $s^l \propto 1 - \frac{n^l + n^{l-1}}{n^l \cdot n^{l-1}}$, where $n^l$ and $n^{l-1}$ represent the number of neurons in layer $l$ and $l-1$ respectively.

For convolutional layers, we use a modification of ER, ERK (Evci et al., 2020), which takes into account the size of the kernels, $s^l \propto 1 - \frac{n^l + n^{l-1} + w^l + h^l}{n^l \cdot n^{l-1} \cdot w^l \cdot h^l}$, where $n^l$ and $n^{l-1}$ represent the number of neurons in layer $l$ and $l-1$ respectively, and $w^l$ and $h^l$ the width and height of the corresponding convolutional kernel.

### 3.2 Dynamic Sparse Training: MagRan-DM and RigL-DM

The key aspect of DST algorithms lies with the process of pruning and regrowing weights. We opted to test the two most common regrowth methods, random growth and gradient growth, combined with the magnitude pruning criteria. Magnitude pruning is a simple criteria, that has been shown to perform well in high sparsity regimes for supervised classification, as well as in other generative models (Nowak et al., 2023; Liu et al., 2023b)

RigL, proposed by Evci et al. (2020), combines gradient growth and magnitude pruning, thus the name of our model RigL-DM. The combination of random growth and magnitude pruning closely resembles the SET algorithm (Mocanu et al., 2018), and has been studied before for other types of models (Nowak et al., 2023), although it has never been named. For simplicity, we refer to this method as MagRan-DM.

---

**Algorithm 2** RigL-DM and MagRan-DM

---

1: **Input:** Dataset $\mathcal{D}$, Network $f_\theta$, Number of Epochs $N$, Diffusion steps $T_d$, Sparsity ratio $S$, exploration frequency $\Delta T_e$, Pruning rate $p$, Sparse method METHOD
2: $\theta \leftarrow$ sparse initialization using $S$
3: **for** $i = 1$ to $N$ **do**
4:      $x_0 \sim \mathcal{D}$
5:      $t \sim \mathcal{U}\left(\{1, 2, \ldots, T_d\}\right)$
6:      $\epsilon \sim \mathcal{N}(\mathbf{0}, \mathbf{I})$
7:      $\theta_i = \text{AdamW}(\nabla_\theta, \mathcal{L}_{\text{DIF}}(f_\theta(x_0, t), \epsilon))$
8:      **if** $i \bmod \Delta T_e$ **then**
9:          $\theta_{i_p} = \text{TopMag}(|\theta_i|, 1 - p)$   // Magnitude pruning
10:          **if** METHOD **is** RigL-DM **then**
11:              $\theta_{i_g} = \text{TopGrad}(|\nabla_\theta \mathcal{L}_{\text{DIF}}|, p)$   // Gradient growth
12:          **else if** METHOD **is** MagRan-DM **then**
13:              $\theta_{i_g} = \text{Random}(p)$   // Random growth
14:          **end if**
15:          $\theta_i \leftarrow$ update activated weights using $\theta_{i_g}$ and $\theta_{i_p}$
16:      **end if**
17: **end for**

---

The full pseudocode for the training process of MagRan-DM and RigL-DM can be found in Algorithm 2. At the start of the training process, the network is sparsely initialized using the same strategy as described for Static-DM. After every $\Delta T_e$ training iterations, a cycle of connection pruning and growth is performed. First, we drop (i.e. set to zero) a fraction of the activated weights with the lowest magnitude from the network, determined using $\text{TopMag}(|\theta_i|, 1 - p)$, which returns the indices of the top $1 - p$ of weights by magnitude. After pruning, we regrow new weights in the same proportion in order to maintain the sparsity level. For RigL-DM, the connections to regrow are given by $\text{TopGrad}(|\nabla_\theta \mathcal{L}_{DIF}|, p)$, that returns the indices of the top $p$ of weights with highest magnitude gradients. For MagRan-DM the regrowth is determined by $\text{Random}(p)$, which outputs the indices of random $p$ of connections.

### 3.3 Experimental Setup

Note that our goal is not to directly compare performance between models or datasets, but to compare the performance of Dense and sparse versions of the same models across different datasets, to gain insights into the impact of sparsity in DM training.

#### 3.3.1 Models and Benchmarks

We test Static-DM, MagRan-DM, and RigL-DM against the Dense baseline, on two different DMs, Latent Diffusion (Rombach et al., 2021) and ChiroDiff (Das et al., 2023), given their popularity among the research literature, on the task of unconditional image generation. Although image generation is the most common application and main direction of current research in DMs, we seek to offer a more extensive look, and

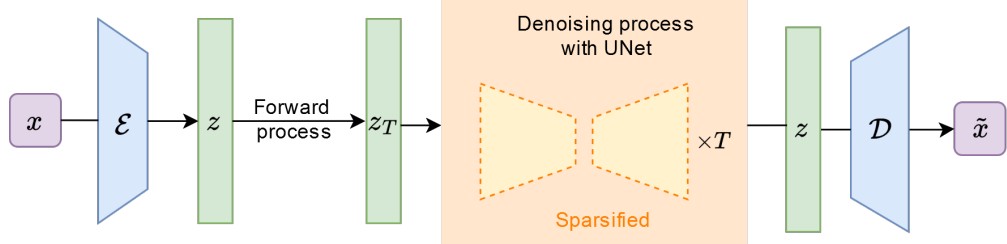

(a) **Latent Diffusion**: Sparsity is applied to the U-Net, leaving the autoencoder parts $(\mathcal{E}, \mathcal{D})$ fully dense.

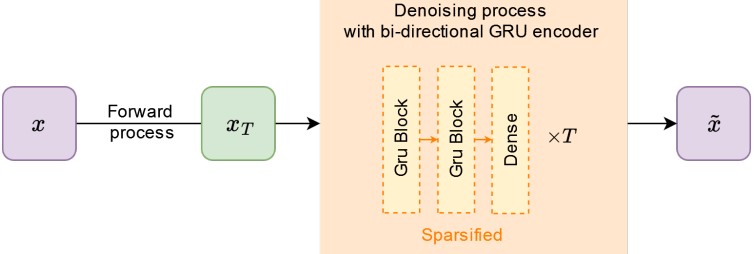

(b) **ChiroDiff**: Sparsity is applied throughout the whole network.

Figure 1: Sparsification of Latent Diffusion (1a) and ChiroDiff (1b) models.

examined DMs for different modalities, with different backbone architectures. More detailed information about the model architectures and choice of datasets can be found in Appendix B.

**Latent Diffusion.**    Latent Diffusion is a DM that creates high-quality images while reducing computational requirements by training in a compressed lower-dimensional latent space. Although we focus on unconditional generation tasks, Latent Diffusion also allows for conditional generation, by using a general-purpose mechanism based on cross attention (Vaswani et al., 2017). Latent Diffusion first employs pre-trained autoencoders to obtain a latent representation of the input, and then performs the diffusion process on these representations, using a U-Net (Ronneberger et al., 2015). Performing the denoising process in the latent space allows to the model to focus on relevant semantic-wise information about the data. We sparsify only the U-Net model, as shown in Figure 1a, and utilize off-the shelf autoencoders provided by Rombach et al. (2021), keeping them dense. We evaluate on the LSUN-Bedrooms (Yu et al., 2015), CelebA-HQ (Karras et al., 2018) and Imagenette (Howard, 2019) datasets.

**ChiroDiff.**    ChiroDiff is a DM specifically designed to model continuous-time chirographic data, such as sketches or handwriting, in the form of a sequence of strokes containing both spatial and temporal information. ChiroDiff can handle sequences of variable length and, as a non-autoregressive model, is able to capture holistic concepts, leading to higher quality samples. This model employs a Bidirectional GRU encoder as backbone architecture. The encoder is fed the spatial coordinates, their point-wise velocities, as well as the entire sequence as context, which provides full context of the sequence during the generation process. Sparsity is applied to the entire network, as shown in Figure 1b. We evaluate it on KanjiVG, QuickDraw (Ha & Eck, 2018), and VMNIST (Das et al., 2022). Following the original paper, we use a preprocessed version of KanjiVG.[1] For QuickDraw we use the following categories: crab, cat, and mosquito; and all results are averaged.

### 3.3.2   Experimental Details

We train the models on a set of sparsity rates $S \in \{0.1, 0.25, 0.5, 0.75, 0.9\}$. For DST methods, we set the exploration frequency $\Delta T_e = 1100$ for all Latent Diffusion datasets, and $\Delta T_e = 800$ for all ChiroDiff datasets.

---

[1]https://github.com/hardmaru/sketch-rnn-datasets/tree/master/kanji

The weight prune and regrowth ratio was set to $p = 0.5$ for all main experiments. These values of $\Delta T_e$ and $p$ were based on a small random search experiment.

Due to computing limitations, we use 12500/500 training/validation images for CelebA-HQ and 10598/2500 images for LSUN-Bedrooms. In Appendix C we conduct experiments using a selection of models with the full CelebA-HQ dataset to demonstrate that using more data does not greatly influence the results. Further, in Appendix D we perform experiments using the ImageNet-1k dataset which contains over 1M images.

To be able to compare the performance of different methods and different sparsity levels, we train the models for a predefined amount of epochs: 150 for Latent Diffusion datasets, and 600 for ChiroDiff datasets. For a complete description of training details please refer to Appendix B.3. Given the extensive number of experiments we conducted, we opted for a shorter training regime. For sampling, we use DDIM sampling (Song et al., 2021) with 100 steps for Latent Diffusion, and 50 steps for ChiroDiff, following the guidance provided in the original papers.

For completeness, we also conducted some experiments on conditional DMs, specifically on class-conditional ImageNet and class-conditional QuickDraw. See Appendix E for details on this setup.

For our experiments, we performed approximately 620 training runs of Dense, Static-DM, RigL-DM, and MagRan-DM models, using two high-performance computer (HPC) clusters equipped with NVIDIA Tesla V100 SXM2 and A100 GPUs. Each DM was trained on only one GPU. All experiments consumed around $6,900$ GPU hours.

### 3.3.3 Evaluation Metrics

We follow common practice and calculate the FID score (Heusel et al., 2017) to assess the performance of all models. Refer to Appendix B.4 for more information on FID calculation. For evaluation completeness, we also report the Kernel Inception Distance (KID) (Bińkowski et al., 2018), with results presented in Appendix I. To evaluate the computational savings of the sparse methods, we report the network size (number of parameters) as a proxy for memory requirement, and the FLOPs, to estimate the computational cost of training and inference. We follow the method of FLOPs calculation described by Evci et al. (2020).

## 4 Experimental Results

We analyze the performance of Static-DM, MagRan-DM, and RigL-DM across various sparsity levels, and compare the results against the original Dense baseline. Additionally, we perform experiments regarding the training dynamics of Dense vs sparse models. Later on, in Section 4.4 we present experiments comparing a selection of DST vs. Dense models across various diffusion timesteps. Examples of the generated samples can be found in Appendix J. Results from the class-conditional experiments are provided in Appendix E.

### 4.1 Latent Diffusion

The results of the studied sparse methods for Latent Diffusion are shown in Figure 2. For CelebA-HQ, 50% of the connections can be removed with minimal to no loss in image quality. With a higher sparsity level of 75%, the three methods still perform comparably to the Dense model, especially Static-DM. However, when the network is very sparse, $S = 0.9$, all models fail to generate high-quality data.

On LSUN-Bedrooms, a similar overall trend can be observed: performance steadily increases with decrease in sparsity level until 25%. Interestingly, MagRan-DM with $S = 0.1$ shows worse performance than the Dense model, and also a significant decrease compared to MagRan-DM with $S = 0.25$. While this goes against the general expectation that more sparsity leads to increasingly worse performance, our intuition is that this might be related to the balance between regularization and expressivenes of the model. When the sparsity is low, the regularization benefits are not very strong, and the model might suffer from a loss of expressiveness due to reduction in parameters, thus obtaining worse results. As such, MagRan-DM with $S = 0.25$ is likely striking a better balance between these two factors. This behaviour can be observed in all three datasets, although less pronounced in CelebA-HQ. However, exploring this topic in depth is beyond the scope of this paper.

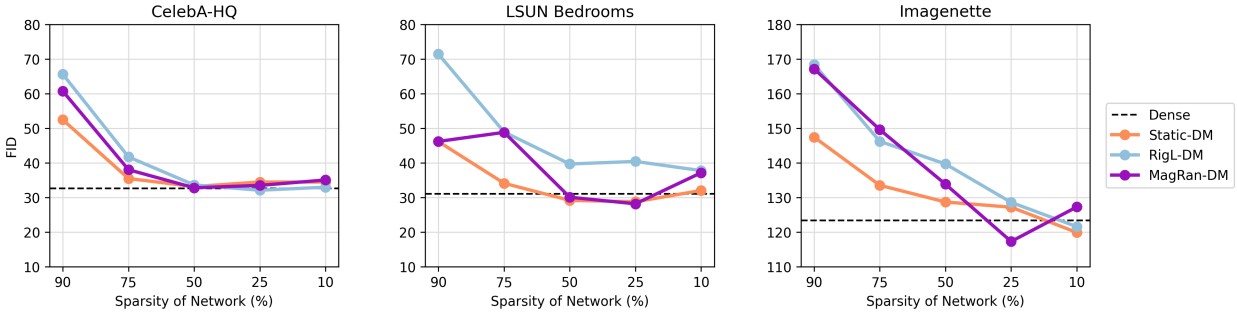

Figure 2: FID score comparisons between Dense, and Static-DM, MagRan-DM and RigL-DM with various sparsity levels, for Latent Diffusion, with prune and regrowth ratio $p = 0.5$. Values are averaged over 3 runs.

Imagenette experiments exhibit the same overall tradeoff between sparsity and performance, with the best results being found in 10% and 25% sparse models.

For all datasets, we successfully trained at least one sparse DM that outperforms the original Dense version. Table 1 presents the metrics for the best sparse models for each method. In CelebA-HQ, only RigL-DM at $S = 0.25$ surpasses Dense performance. In LSUN-Bedrooms, both Static-DM and MagRan-DM were able to outperform it. In Imagenette, all methods were able to achieve superior performance, albeit at different sparsity levels. We note that the variance observed in the models is similar when comparing dense and sparse versions in all cases.

Table 1: Performance and cost of training and testing of Dense and best Static-DM, RigL-DM, and MagRan-DM versions for Latent Diffusion. Values are averaged over 3 runs. The FLOPs of sparse DMs are normalized with the FLOPs of their Dense versions. Test FLOPS were calculated for one sample. Sparse models that outperform the Dense version are marked in bold. The top-performing sparse model is underlined.

| Dataset | Approach | FID $\pm$ SD ($\downarrow$) | Params | Train FLOPs | Test FLOPs |
|---|---|---|---|---|---|
| CelebA-HQ | Dense | 32.74 $\pm$ 3.68 | 274.1M | 9.00e16 | 1.92e13 |
| | Static-DM, $S = 0.5$ | 33.19 $\pm$ 2.39 | 0.50$\times$ | 0.68$\times$ | 0.68$\times$ |
| | RigL-DM, $S = 0.25$ | **32.12 $\pm$ 3.10** | 0.75$\times$ | 0.91$\times$ | 0.91$\times$ |
| | MagRan-DM, $S = 0.5$ | 32.83 $\pm$ 1.68 | 0.50$\times$ | 0.67$\times$ | 0.67$\times$ |
| Bedrooms | Dense | 31.09 $\pm$ 12.42 | 274.1M | 7.64e16 | 1.92e13 |
| | Static-DM, $S = 0.25$ | **28.79 $\pm$ 12.65** | 0.75$\times$ | 0.91$\times$ | 0.91$\times$ |
| | RigL-DM, $S = 0.10$ | 37.80 $\pm$ 13.55 | 0.90$\times$ | 0.97$\times$ | 0.97$\times$ |
| | MagRan-DM, $S = 0.25$ | **28.20 $\pm$ 7.64** | 0.75$\times$ | 0.91$\times$ | 0.91$\times$ |
| Imagenette | Dense | 123.42 $\pm$ 4.25 | 274.1M | 6.83e16 | 1.92e13 |
| | Static-DM, $S = 0.10$ | **119.92 $\pm$ 5.94** | 0.90$\times$ | 0.97$\times$ | 0.97$\times$ |
| | RigL-DM, $S = 0.10$ | **121.59 $\pm$ 6.91** | 0.90$\times$ | 0.97$\times$ | 0.97$\times$ |
| | MagRan-DM, $S = 0.25$ | **117.32 $\pm$ 8.52** | 0.75$\times$ | 0.91$\times$ | 0.91$\times$ |

**Memory and computational savings.** In Table 1, we can observe that the top-performing sparse DM on CelebA-HQ, RigL-DM with $S = 0.25$, is able to outperform Dense performance, while reducing by 25% the number of parameters and 10% the number of FLOPs. Although Static-DM $S = 0.5$ and MagRan-DM $S = 0.5$ achieve slightly inferior performance, they are able reduce FLOPS and number of parameters more significantly, by 30% and 50%, respectively. On LSUN-Bedrooms and Imagenette, the top-performing sparse DM reduces number of FLOPs by 10%, and number of parameters by 25%.

**Prune and regrowth rate experiments.** In all datasets, Static-DM has better performance than the dynamic methods in higher sparsity setups, $S > 0.5$. This is interesting, as it departs from the usual patterns

found in sparse-to-sparse training for supervised learning applications and even other generative models such as GANs, where DST usually outperforms SST (Mocanu et al., 2018; Liu et al., 2023b). Liu et al. (2021c) found that, in image classification tasks, DST models consistently achieve better performance over SST with appropriate parameter exploration, i.e., exploration frequency $\Delta T_e$ and prune and regrowth ratio $p$. To provide insights on the importance of $p$ for DST experiments, we conducted an experiment using a prune and regrowth rate $p \in \{0.05, 0.1, 0.2, 0.3, 0.5\}$. The results are provided in Figure 8 in Appendix H. The best results were obtained with $p = 0.05$.

Following this experiment, we repeated all experiments for DST methods presented in Figure 2, using $p = 0.05$, and show the results in Figure 9 and Appendix H. One particularly interesting finding is that, in high sparsity regimes, such as $S = 0.9$ and $S = 0.75$, DST methods have consistently better performance when $p = 0.05$, even outperforming Static-DM. However, this performance advantage disappears when using the more aggressive prune and regrowth rate of $p = 0.5$. Please refer to Appendix H for a more in-depth analysis.

### 4.2 ChiroDiff

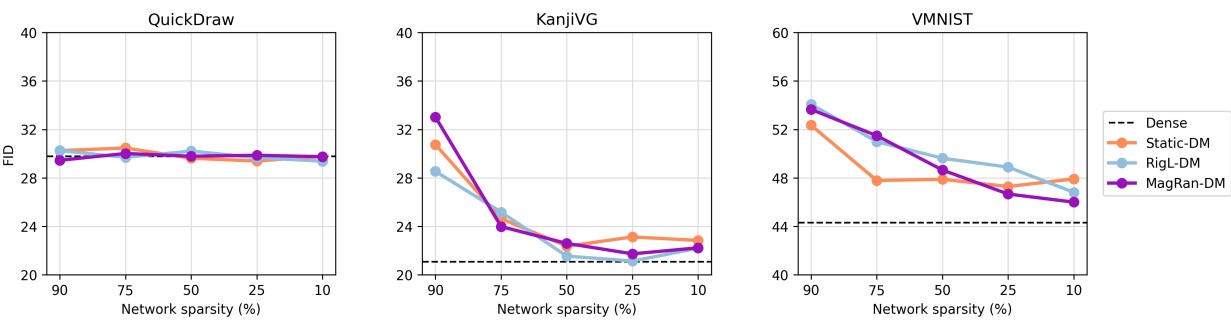

Figure 3: FID score comparisons between Dense, and Static-DM, MagRan-DM and RigL-DM with various sparsity levels, for ChiroDiff, with prune and regrowth rate $p = 0.5$. Values averaged over 3 runs.

Figure 3 shows the FID scores of the studied sparse methods for ChiroDiff. For QuickDraw, we observe that both Static-DM and RigL-DM exhibit variations around the performance of the Dense model, with only a subtle tendency to deteriorate as sparsity increases. MagRan-DM consistently matches the FID of the Dense model, and is able to outperform it at 90% sparsity. These results suggest that this model is overparameterized, which would explain why it benefits significantly from sparsity, even when removing 90% of the weights. We repeated the same experiments using a smaller version of the model, the results of which are shown in Appendix F. Although most sparse models still perform comparably to the dense version, the trend of diminishing performance with increased sparsity is apparent, in contrast to the results observed with the larger model.

On KanjiVG, the impact of sparsity is more pronounced, as all three methods demonstrate a downward trend in performance as sparsity increases. Dynamic methods have consistently better performance than Static-DM, and RigL-DM exhibits top performance in all sparsity levels except for $S = 0.75$.

In VMNIST experiments, there is, again, a pattern of better performance as sparsity decreases. Similarly to Latent Diffusion experiments, SST has better performance in higher sparsity settings, $S > 0.5$. In this dataset, there is a slighter larger gap in performance between the sparse and dense models.

We successfully trained at least one sparse DM from each method that demonstrates a comparable performance to the Dense counterpart, and show the results on Table 2. RigL-DM was the top-performing method on QuickDraw, with $S = 0.1$, and on KanjiVG, with $S = 0.25$, while in VMNIST, the top method was MagRan-DM, with $S = 0.10$. For QuickDraw, the top sparse DM was able to outperform the Dense network.

**Memory and computational savings.** Table 2 shows that the top-performing sparse DM on KanjiVG achieves a reduction in the number of parameters and FLOPs of about 30%, while achieving a similar FID score. On Quickdraw, MagRan-DM with 90% sparsity achieves an considerable reduction of 88%, and even

Table 2: Performance and cost of training and testing of the Dense and best Static-DM, RigL-DM, and MagRan-DM for ChiroDiff. Values averaged over 3 runs. The FLOPs of sparse DMs are normalized with the FLOPs of the dense versions, and test FLOPS were calculated for one sample. Sparse models that outperform the Dense version are marked in bold. The top-performing sparse model is underlined.

| Dataset | Approach | FID ± SD (↓) | Params | Train FLOPs | Test FLOPs |
|---|---|---|---|---|---|
| QuickDraw | Dense | 29.78 ± 0.59 | 736027 | 5.12 e14 | 1.29 e10 |
| | Static-DM, $S = 0.25$ | **29.39 ± 0.24** | 0.75× | 0.75× | 0.75× |
| | RigL-DM, $S = 0.10$ | **29.38 ± 0.27** | 0.89× | 0.89× | 0.89× |
| | MagRan-DM, $S = 0.90$ | **29.45 ± 0.39** | 0.10× | 0.10× | 0.10× |
| KanjiVG | Dense | 21.10 ± 0.25 | 416859 | 1.80e13 | 7.35 e9 |
| | Static-DM, $S = 0.5$ | 22.36 ± 0.87 | 0.50× | 0.51× | 0.51× |
| | RigL-DM, $S = 0.25$ | 21.14 ± 0.71 | 0.70× | 0.70× | 0.70× |
| | MagRan-DM, $S = 0.25$ | 21.73 ± 1.18 | 0.39× | 0.39× | 0.39× |
| VMNIST | Dense | 44.21 ± 0.62 | 65019 | 1.69e12 | 7.11 e8 |
| | Static-DM, $S = 0.25$ | 47.29 ± 1.96 | 0.75× | 0.74× | 0.74× |
| | RigL-DM, $S = 0.10$ | 46.81 ± 1.98 | 0.90× | 0.90× | 0.89× |
| | MagRan-DM, $S = 0.10$ | 46.00 ±1.71 | 0.90× | 0.89× | 0.89× |

though it is not the top-performing sparse model, it also outperforms the Dense model. The top sparse model on VMNIST, provides a reduction in FLOPs of about 89%.

**Prune and regrowth rate experiments.** Similar to Latent Diffusion, we repeated the DST experiments using the more conservative prune and regrowth rate of 0.05. The biggest improvement was seen in the Quickdraw dataset, where DST methods obtained considerably higher performances, as compared with Figure 3. Akin to the Latent Diffusion results, in higher sparsity regimes DST models mostly obtain better performance when using $p = 0.05$. Please refer to Appendix H for a more in-depth analysis.

## 4.3 Training Dynamics

In addition to final FID scores, we analysed training dynamics to better understand the behaviour of sparse models during training. In order to perform this analysis, we selected the sparsity level that achieved the overall best results in previous experiments, $S = 0.25$, and plotted the FID scores across several epochs.

In general, sparse models appear to follow the trend of the corresponding dense model, which appears to indicate that they retain the stable training behaviour of dense diffusion models.

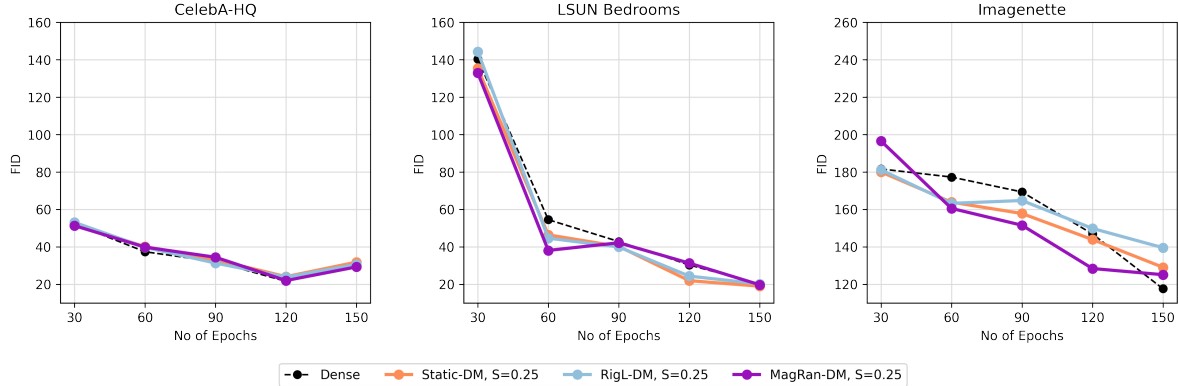

Figure 4: Comparison of training dynamics between Dense, and Static-DM, MagRan-DM and RigL-DM with $S = 0.25$, for Latent Diffusion.

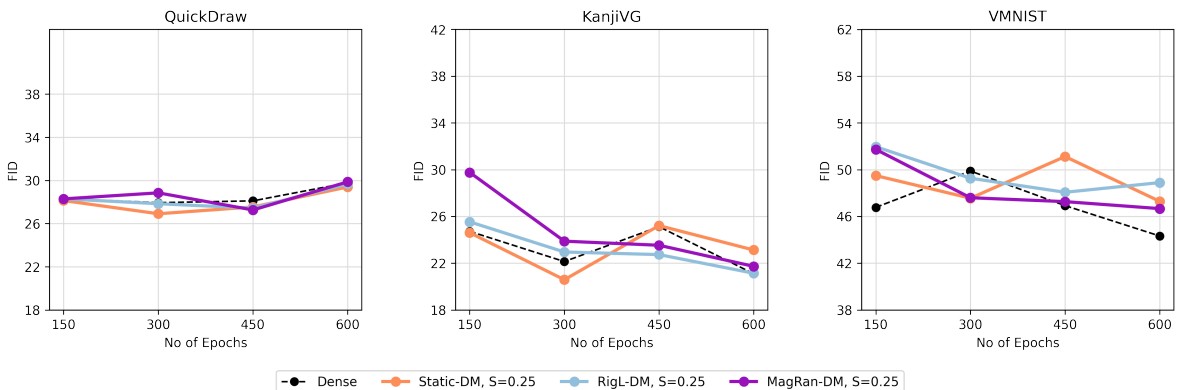

Figure 5: Comparison of training dynamics between Dense, and Static-DM, MagRan-DM and RigL-DM with $S = 0.25$, for ChiroDiff.

## 4.4 Impact of Diffusion Steps

The number of timesteps is an important parameter in DMs, as too few can lead to insufficient denoising, and low quality images, while too many might increase computational complexity without improving output quality. We explored the relationship between the number of timesteps and model sparsity, aiming to determine whether a very sparse model ($S = 0.75$) with an increased number of sampling steps can achieve performance comparable to that of a dense model, with less sampling steps. We perform experiments using CelebA-HQ for Latent Diffusion, and KanjiVG for ChiroDiff, the results of which are presented in Figure 6.

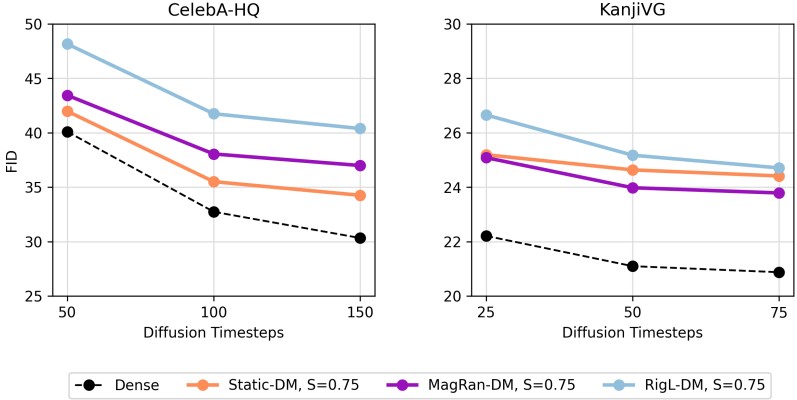

Figure 6: FID score comparisons between Dense, and Static-DM, MagRan-DM and RigL-DM with $S = 0.75$, using varied diffusion timesteps for Latent Diffusion (CelebA-HQ), and ChiroDiff (KanjiVG). Values averaged over 3 runs.

In general, the number of sampling steps does not affect when comparing sparse and dense versions within the same number of timesteps. More experiments are presented in Appendix G. In KanjiVG, no sparse model is able to match any version of the dense model, and varying the number of timesteps appears to have little influence on the quality of the output. In CelebA-HQ, when comparing different numbers of timesteps, we observe that MagRan-DM and Static-DM with both 100 and 150 timesteps are able to outperform the Dense model using 50 timesteps. As an example, in Static-DM, $S = 0.75$ with 100 timesteps vs. the dense model with 50 timesteps, Static-DM offers a theoretical speedup of $0.29\times$ over the dense model's Training FLOPs, and $0.57\times$ of the Testing FLOPs, while creating better quality samples.

### 4.5 Limitations and Future Work

Apart from the previously mentioned computational limitations when training on CelebA-HQ and LSUN-Bedrooms, our findings demonstrate systematic trends that prompt for further investigation. Training for longer epochs could provide deeper insights into the capabilities of sparse models. Additionally, there is potential in exploring other pruning strategies and other DST hyperparameters such as $\Delta T_e$. Another interesting direction is to adjust DST hyperparameters based on the training phase, in response to changes in training dynamics. Furthermore, employing multiple sparsity masks with varying sparsity levels and dynamically changing them during training, according to the denoising timestep, is a promising line of research.

Table 3: Overview of dense vs sparse Latent Diffusion (LD) and ChiroDiff (CD) models. Experiments where the sparse model outperforms the dense version are marked in grey.

| Model | Dataset | Dense FID | Best Sparse model Model $(S, p)$ | FID | FLOPs Train & Test |
|-------|---------|-----------|----------------------------------|-----|--------------------|
| LD | CelebA-HQ | 32.74 | RigL-DM (25%, 0.5) | 32.12 | 0.91× |
| LD | Bedrooms | 31.09 | MagRan-DM (10%, 0.05) | 25.12 | 0.97× |
| LD | Imagenette | 123.42 | MagRan-DM (25%, 0.5) | 117.32 | 0.91× |
| | | | | | |
| CD | QuickDraw | 29.78 | RigL-DM (25%, 0.05) | 24.91 | 0.75× |
| CD | Kanji-VG | 21.10 | MagRan-DM (50%, 0.05) | 20.32 | 0.51× |
| CD | VMNIST | 44.21 | MagRan-DM (10%, 0.5) | 46.00 | 0.89× |

## 5 Conclusion

We have introduced sparse-to-sparse training of DMs. Our experiments show that both SST and DST methods are able to match and often outperform the dense DMs, as shown in Table 3, while reducing memory and computational costs. We highlight the importance of choosing the correct method and sparsity level, depending on the model (and even the dataset) that is being used. Taken together, our findings show the great potential of sparse-to-sparse training in improving the efficiency of both training and sampling from DMs.

**Open Science:** Our code and models are available at `https://github.com/iclbo/sparse_to_sparse_diffusion`

### Acknowledgements

Research supported by Fonds National de la Recherche Luxembourg - FNR (SCRIPTOR project, grant AFR/22/17177001) and the European Innovation Council Pathfinder program (SYMBIOTIK project, grant 101071147).

The experiments presented in this paper were carried out using the HPC facilities of the University of Luxembourg (`https://hpc.uni.lu`) and Luxembourg's national supercomputer MeluXina. The authors gratefully acknowledge the ULHPC and LuxProvide teams for their expert support.

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

# A    Hardware and Software Support

One of the main challenges in sparse neural networks research is that most hardware optimized for deep learning is designed for dense matrix operations. As a result, most of current research attempts to mimic sparsity by using a binary mask over weights, which results in sparse networks offering, in practice, no better training efficiency than dense networks. However, industry is catching up and so it is a matter of time for hardware to truly leverage sparse operations.

There is a growing trend towards developing hardware that better supports sparse operations. In 2021, NVIDIA released the A100 GPU, which supports accelerating operations in a 2:4 sparsity pattern. Several works have already leveraged this feature (Zhou et al., 2021; Wang et al., 2024). In order to use this capability, the sparse matrices must follow a specific structure: among each group of four contiguous values, two values must be zero, thereby fixing the sparsity level at 50%. While this structure enables significant acceleration, it supports only one static sparsity level, and makes it impossible to vary the sparsity ratio between layers.

More recently, Cerebras introduced the CS-3 AI accelerator (Lie, 2022), capable of accelerating sparse training and supporting unstructured sparsity. Using Cerebras' CS-3 AI to accelerate training, and Neural Magic's inference server to accelerate inference, Agarwalla et al. (2024) trained an accurate sparse Llama-2 7B model. Its accelerated training closely matched the theoretical speedup, while achieving 91.8% accuracy recovery of Llama Evaluation metrics, with 70% sparsity. This significant finding underscores the potential of sparse training to produce more efficient neural networks in practice, not just in theory.

In parallel, there have also been advancements in creating software implementations that support truly sparse-to-sparse neural network training, mostly for supervised learning tasks (Liu et al., 2021b; Curci et al., 2022). In addition, a sparse-to-sparse denoising autoencoder has been developed by Atashgahi et al. (2022), to perform fast and robust feature selection.

These developments in both hardware and software point towards a future where sparse-to-sparse training may become the de facto approach for developing neural networks, enabling faster, more memory-efficient, and energy-efficient deep learning models.

# B    Experiments setup

## B.1    Model Architectures

In Latent Diffusion experiments, the model architecture is the same for LSUN-Bedrooms, CelebA-HQ and Imagenette datasets. The DM follows the architecture proposed by Rombach et al. (2021). For the autoconder, we utilize a pre-trained model released by the Latent Diffusion authors on the project's GitHub,[2] with spatial size 64x64x3, VQ-reg regularization, and downsampling factor $f = 4$.

In ChiroDiff experiments, we adopt the architecture proposed by Das et al. (2023). The backbone network is a bidirectional GRU encoder with 3 layers, with 96 hidden units for KanjiVG, and 128 hidden units for QuickDraw. For VMNIST, the backbone network is a 2-layer bidirectional GRU encoder with 48 hidden units. We also use the code available on the project's GitHub repository.[3]

## B.2    Choice of Datasets

We evaluated Latent Diffusion on LSUN-Bedrooms and CelebA-HQ, following their use in the original paper. Additionally, we included Imagenette, a subset of the popular ImageNet (Deng et al., 2009) dataset. For ChiroDiff, we used the same datasets evaluated as the original study: QuickDraw, KanjiVG and VMNIST. While the authors of ChiroDiff analysed seven categories of QuickDraw, namely {cat, crab, mosquito, bus, fish, yoga, flower}, we opted to reduce the number of categories to {cat, crab, mosquito} given the large number of experiments involved in our investigation. In Appendix C below we demonstrate that dataset

---

[2]https://github.com/CompVis/latent-diffusion
[3]https://github.com/dasayan05/chirodiff

size does not change the main outcomes. Ultimately, our goal is to compare and contrast sparse and dense models, independent of dataset size.

### B.3 Training Regime

We follow the configurations provided in the GitHub repositories of the original papers and present the main aspects below. The only alterations made were in the batch size and learning rate. The only exception is Imagenette, which was not included in the original paper; for this dataset, we applied the same configuration settings as those used for LSUN-Bedrooms.

**Latent Diffusion on LSUN-Bedrooms**: We use a batch size of 12, AdamW optimizer with weight decay 1e-2 and static learning rate 2.4e-5. We train for 150 epochs. We use 1000 Denoising steps (T), linear noise schedule from 0.0015 to 0.0195, and sinusoidal embeddings for the timestep.

**Latent Diffusion on CelebA-HQ**: We use a batch size of 12, AdamW optimizer with weight decay 1e-2 and static learning rate 2.0e-06. We train for 150 epochs. We use 1000 Denoising steps (T), linear noise schedule from 0.0015 to 0.0195, and sinusoidal embeddings for the timestep.

**Latent Diffusion on Imagenette**: We use a batch size of 12, AdamW optimizer with weight decay 1e-2 and static learning rate 2.4e-5. We train for 150 epochs. We use 1000 Denoising steps (T), linear noise schedule from 0.0015 to 0.0195, and sinusoidal embeddings for the timestep.

**ChiroDiff on QuickDraw**: We use a batch size of 128, AdamW optimizer with weight decay 1e-2 and static learning rate 1e-3. We train for 600 epochs. We use 1000 Denoising steps (T), linear noise schedule from 1e-4 to 2e-2, and random Fourier features for the timestep embedding.

**ChiroDiff on KanjiVG**: We use a batch size of 128, AdamW optimizer with weight decay 1e-2 and static learning rate 1e-3. We train for 600 epochs. We use 1000 Denoising steps (T), linear noise schedule from 1e-4 to 2e-2, and random Fourier features for the timestep embedding.

**ChiroDiff on VMNIST**: We use a batch size of 128, AdamW optimizer with weight decay 1e-2 and static learning rate 1e-3. We train for 600 epochs. We use 1000 Denoising steps (T), linear noise schedule from 1e-4 to 2e-2, and random Fourier features for the timestep embedding.

Each setup was trained for 5 sparsity values $[0.1, 0.25, 0.5, 0.75, 0.9]$, and we perform 3 runs for each model/dataset/sparsity combination. For ChiroDiff on QuickDraw, we trained each category {cat, crab, mosquito} for 3 runs, resulting in a total of 9 runs per sparsity level.

### B.4 FID calculation

For Latent Diffusion, FID is calculated using the `torch-fidelity` Python package, and estimated based on 10k samples and the entire training set, as in the original work. For ChiroDiff, following the original paper, we plot and save the chirographic sequences as images, and calculate the FID using the inception model provided by Ge et al. (2020), pre-trained on the QuickDraw dataset, using 10k generated samples and 20k real samples.

## C   Experiments using the Full CelebA-HQ Dataset

We conducted experiments using Static-DM, MagRan-DM, and RigL-DM with $S = 0.5$ on the full CelebA-HQ dataset, for 150 epochs, and compare the results with the previous models trained on 50% of the dataset. As shown in Table 4, the FID scores are similar across both datasets for each respective method. This supports our decision to focus on a subset of the dataset for our main experiments, to save valuable computational resources. Interestingly, all sparse models are able to outperform their dense version when trained on the full dataset.

Table 4: Comparison of FID and KID scores for Latent Diffusion on CelebA-HQ using full dataset vs. reduced dataset. Results are based on the first run. Sparse models that outperform their Dense version are marked in bold. The top-performing sparse model is underlined.

| Methods | FID (↓) | | KID (↓) | |
|---|---|---|---|---|
| | Full dataset | Reduced dataset | Full dataset | Reduced dataset |
| Dense | 32.20 | 29.68 | 0.0259 | 0.0241 |
| Static-DM, $S = 0.50$ | **29.71** | 29.91 | **0.0242** | 0.0243 |
| RigL-DM, $S = 0.50$ | **30.98** | 30.82 | **0.0250** | 0.0254 |
| MagRan-DM, $S = 0.50$ | **26.70** | 30.71 | **0.0208** | 0.0253 |

## D Experiments using the Full ImageNet-1k Dataset

In this section, we present the results for the most promising sparse DM trained with the ImageNet-1k dataset, comprising 1000 classes spanning $1,281,167$ training images, $50,000$ validation images and $100,000$ test images.

Table 5: FID and KID scores for Latent Diffusion on ImageNet.

| Methods | FID (↓) | KID (↓) |
|---|---|---|
| Dense | 63.95 | 0.0538 |
| MagRan-DM, $S = 0.50$ | 77.39 | 0.0714 |

## E Experiments using Conditional Models

While the main focus of this paper is investigating how sparse-to-sparse training affects unconditional models, we also performed some initial experiments on conditional models to explore its applicability in this setting. Specifically we conducted experiments on class-conditional ImageNet using Latent Diffusion, and class-conditional QuickDraw using ChiroDiff.

For these experiments, models were trained for 50 epochs on ImageNet and 150 on Quickdraw. For ImageNet, all classes were utilized in training and sampling was performed using $\eta$=1.0 and a classifier-free guidance scale of 3.0. For QuickDraw, seven classes were used: cat, crab, mosquito, bus, flower, yoga and fish. Examples of generated samples can be found in Figure 21 and Figure 22 in Appendix J.

On class-conditional ImageNet, the dense model outperforms the sparse version, with a moderate gap between them. On Quickdraw, models show extremely similar performance, with the best results achieved by MagRan-DM with 50% sparsity. These results suggest that sparse-to-sparse training may also be effective for conditional models.

Table 6: FID and KID scores for Latent Diffusion on class-conditional ImageNet.

| Methods | FID (↓) | KID (↓) |
|---|---|---|
| Dense | 17.33 | 0.0071 |
| MagRan-DM, $S = 0.50$ | 20.33 | 0.0088 |

Table 7: FID and KID scores for ChiroDiff on class-conditional QuickDraw. Sparse models that outperform their Dense version are marked in bold. The top-performing sparse model is underlined.

| Methods | FID ($\downarrow$) | KID ($\downarrow$) |
|---|---|---|
| Dense | 31.23 | 0.0327 |
| Static-DM, $S = 0.50$ | 31.57 | **0.0310** |
| MagRan-DM, $S = 0.50$ | **30.20** | **0.0300** |
| RigL-DM, $S = 0.50$ | 32.12 | **0.0315** |

## F    Experiments using a smaller model

In order to explore the strong results of high sparsity models ($S = 0.90$) on QuickDraw using ChiroDiff, we repeated the experiments in Figure 3, using a smaller model. In this smaller model, the backbone network is a bidirectional GRU encoder consisting of 3 layers with 96 hidden units each, compared to 128 in the larger model.

In contrast with the larger model, the trend of diminishing performance as sparsity increases, present in the other datasets, can be observed. Most sparse models perform comparably to the dense, although some performance degradation is apparent in high sparsity models.

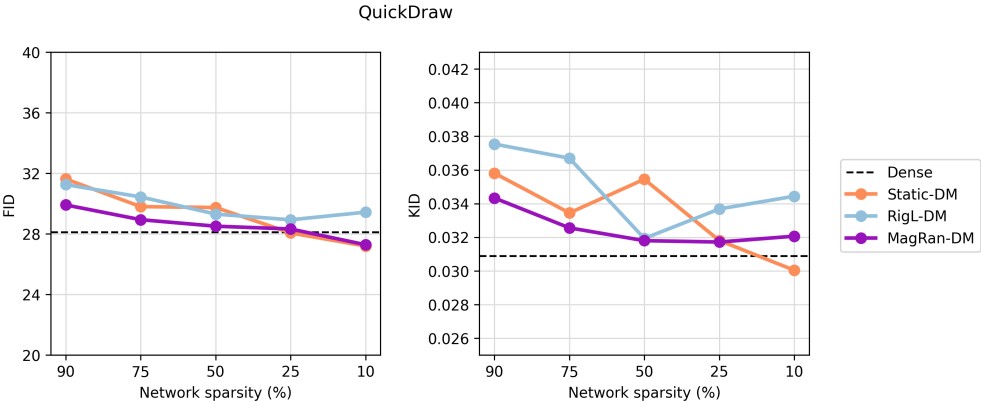

Figure 7: FID and KID scores comparisons between Dense, and Static-DM, MagRan-DM and RigL-DM with various sparsity levels, for QuickDraw dataset, with prune and regrowth rate p = 0.5.

## G    Experiments using Various Diffusion Timesteps

In Table 8, we report the results of the models listed in Table 1 using 50, 100 and 200 sampling steps. These experiments confirm that the number of sampling steps typically does not affect whether a sparse model outperforms a dense model. In other words, a sparse model that performs better than a dense model at 100 timesteps also outperforms it at 50 and 200 timesteps.

For CelebA-HQ, the variation in timesteps does not change the top-performing model, which is consistently RigL-DM with $S = 0.25$. However, in LSUN-Bedrooms, the top-performing method varies with different timesteps.

## H    Prune and Regrowth Rate Experiments

To provide insight on the importance of the prune and regrowth rate for DST experiments, we conducted an experiment using varying values, with the top MagRan-DM and RigL-DM models for the CelebA-HQ

Table 8: Comparison of FID scores for models listed in Table 1 using various DDIM sampling steps. Results based on the first run. Sparse models that outperform the Dense version, in the respective sampling steps, are marked in bold. The top-performing sparse model for each sampling step is underlined.

| **Dataset** | | **FID** ($\downarrow$) | | |
|---|---|---|---|---|
| | | 50 steps | 100 steps | 200 steps |
| CelebA-HQ | Dense | 38.14 | 29.68 | 26.47 |
| | Static-DM, $S = 0.5$ | 38.16 | 29.91 | **26.44** |
| | RigL-DM, $S = 0.25$ | **36.55** | **28.00** | **25.25** |
| | MagRan-DM, $S = 0.5$ | 39.31 | 30.71 | 28.02 |
| LSUN-Bedrooms | Dense | 20.42 | 20.14 | 20.58 |
| | Static-DM, $S = 0.25$ | **20.01** | **18.96** | **19.26** |
| | RigL-DM, $S = 0.10$ | **19.01** | **17.96** | 20.49 |
| | MagRan-DM, $S = 0.25$ | 20.69 | **18.23** | **17.87** |

dataset, listed in Table 1. We report the results of the experiments in Figure 8. Although all FID values are extremely similar, the best performing models for both algorithms use prune and regrowth ratio $p = 0.05$, and both outperform the Dense version. This suggests that selecting an optimal ratio can improve model performance, even if only slightly in these lower-sparsity models tested.

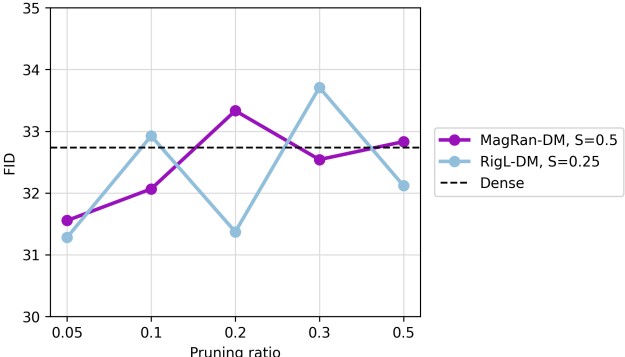

Figure 8: FID scores comparison between Dense and DST models with various prune and regrowth ratios, for Latent Diffusion on CelebA-HQ.

Informed by the results of Figure 8, we conducted experiments for DST methods using the same setup as in Figure 2 and Figure 3, but using a prune and regrowth rate of $p = 0.05$. As can be observed in Figure 9 and Table 9, the general trend of diminishing performance when sparsity increases still remains, with the exception of QuickDraw, in which all DST models had a significant increase in performance.

When looking at very high sparsity regimes, $S = 0.90$, we observe that most models continue to suffer from a significant performance drop when compared to their Dense version, except Quickdraw, where the new prune and regrowth provides a remarkable improvement, and LSUN Bedrooms, where MagRan-DM has an impressively high performance. However, when $S = 0.90$ (and to a lower degree $S = 0.75$) DST methods using $p = 0.05$ have consistently better performance than their $p = 0.5$ counterparts. When comparing the DST methods to Static-DM at $S = 0.90$, in Table 9, we observe that at least one DST method is able to outperform Static-DM in CelebA-HQ and LSUN-Bedrooms, or closely match it in Imagenette, which did not happen with $p = 0.5$. Similarly, for ChiroDiff, in Table 10, almost all DST methods in all three datasets are able to outperform Static-DM.

All in all, these findings suggest that a prune and regrowth ratio of 0.5 is too aggressive, and that a more conservative choice of 0.05 is more appropriate for DMs. Previous work has mentioned that DST methods

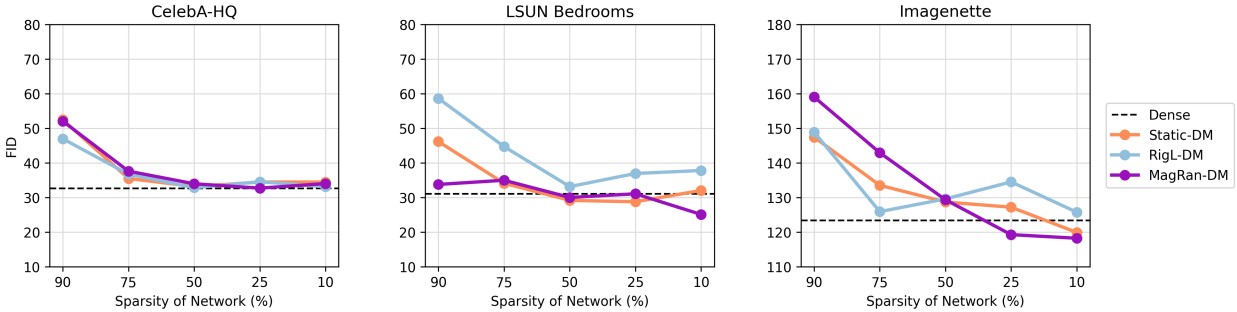

Figure 9: FID score comparisons between Dense and Sparse versions (Static-DM, MagRan-DM, RigL-DM) considering various sparsity levels for Latent Diffusion. DST method use a prune and regrowth rate of 0.05. Values averaged over 3 runs.

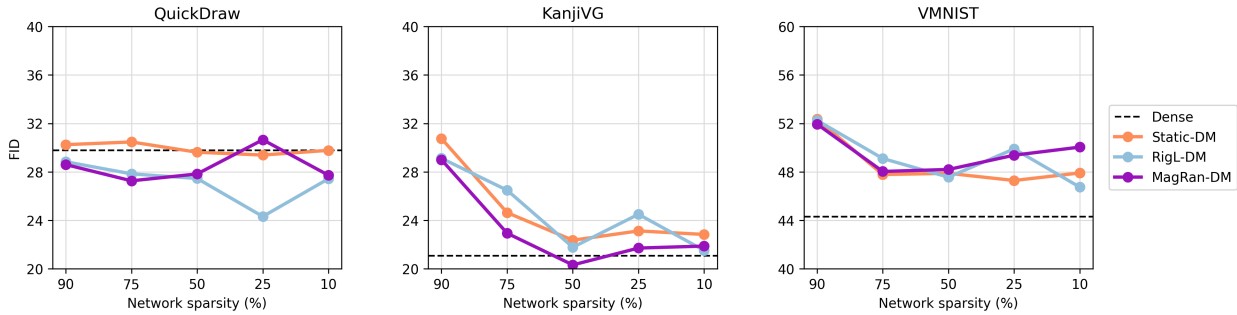

Figure 10: FID score comparisons between Dense and Sparse versions (Static-DM, MagRan-DM, RigL-DM) considering various sparsity levels for ChiroDiff. DST method use a prune and regrowth rate of 0.05. Values averaged over 3 runs.

Table 9: Comparison of FID scores for SST (Static-DM) and DST (RigL-DM, MagRan-DM) models, with $S = 0.9$ using two different prune and regrowth rates ($p = 0.5$ and $p = 0.05$) for Latent Diffusion. DST models that outperform SST are marked in bold.

| Dataset | Static-DM | RigL-DM $p = 0.5, p = 0.05$ | MagRan-DM $p = 0.5, p = 0.05$ |
|---|---|---|---|
| CelebA-HQ | $52.48 \pm 4.88$ | $65.65 \pm 4.32$, $\mathbf{46.07 \pm 11.08}$ | $60.77 \pm 6.58$, $\mathbf{48.39 \pm 14.05}$ |
| Bedrooms | $46.18 \pm 13.42$ | $71.45 \pm 18.84$, $58.64 \pm 22.88$ | $46.22 \pm 10.11$, $\mathbf{33.80 \pm 3.98}$ |
| Imagenette | $147.47 \pm 7.74$ | $168.48 \pm 15.15$, $148.93 \pm 12.03$ | $167.19 \pm 8.20$, $159.08 \pm 14.68$ |

Table 10: Comparison of FID scores for SST (Static-DM) and DST (RigL-DM, MagRan-DM) models, with $S = 0.9$ using two different prune and regrowth rates ($p = 0.5$ and $p = 0.05$) for ChiroDiff. DST models that outperform SST are marked in bold.

| Dataset | Static-DM | RigL-DM $p = 0.5, p = 0.05$ | MagRan-DM $p = 0.5, p = 0.05$ |
|---|---|---|---|
| QuickDraw | $30.25 \pm 0.43$ | $30.26 \pm 0.63$, $\mathbf{28.84 \pm 0.37}$ | $\mathbf{29.45 \pm 0.39}$, $\mathbf{28.60 \pm 0.37}$ |
| KanjiVG | $30.75 \pm 2.16$ | $\mathbf{28.54 \pm 0.74}$, $\mathbf{29.12 \pm 0.57}$ | $33.02 \pm 3.28$, $\mathbf{29.01 \pm 1.48}$ |
| VMNIST | $52.35 \pm 0.84$ | $54.08 \pm 1.57$, $\mathbf{52.25 \pm 0.20}$ | $53.65 \pm 0.69$, $\mathbf{51.94 \pm 1.12}$ |

are consistently superior to SST as long as there is appropriate parameter exploration (Liu et al., 2021c), an observations that aligns with our findings.

# I   Additional evaluation metrics

In this section, we present the KID results corresponding to previously reported experiments, in Figure 2, Figure 3, Figure 9 and Figure 10.

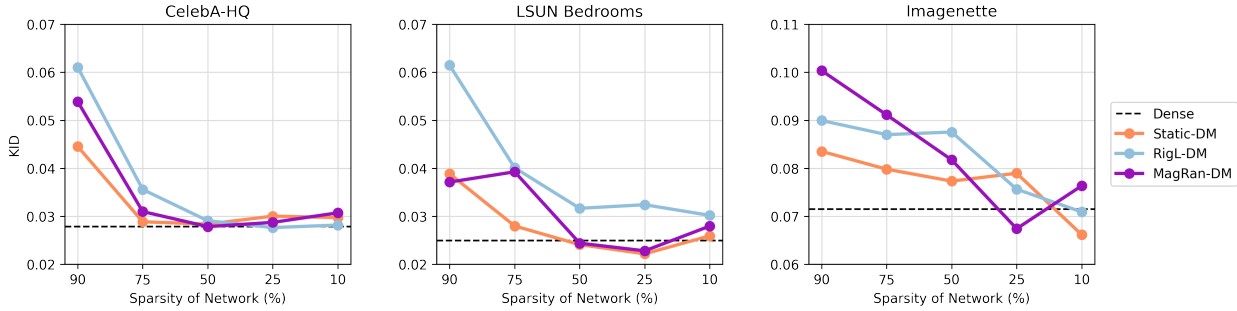

Figure 11: KID comparisons between Dense, and Static-DM, MagRan-DM and RigL-DM with various sparsity levels, for LatentDiffusion, with prune and regrowth rate $p = 0.5$.

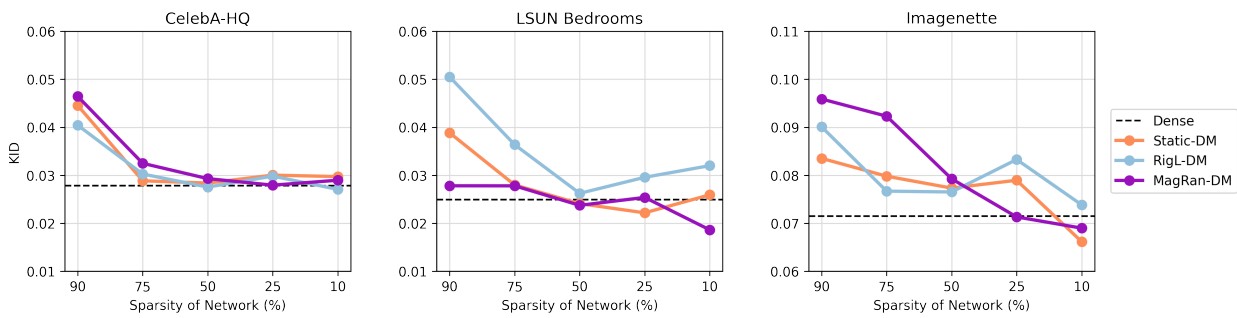

Figure 12: KID comparisons between Dense, and Static-DM, MagRan-DM and RigL-DM with various sparsity levels, for LatentDiffusion, with prune and regrowth rate $p = 0.05$.

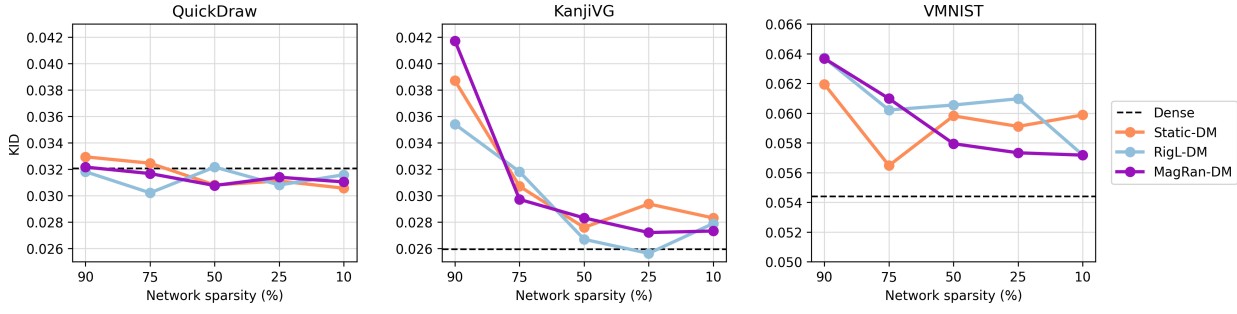

Figure 13: KID comparisons between Dense, and Static-DM, MagRan-DM and RigL-DM with various sparsity levels, for ChiroDiff, with prune and regrowth rate $p = 0.5$.

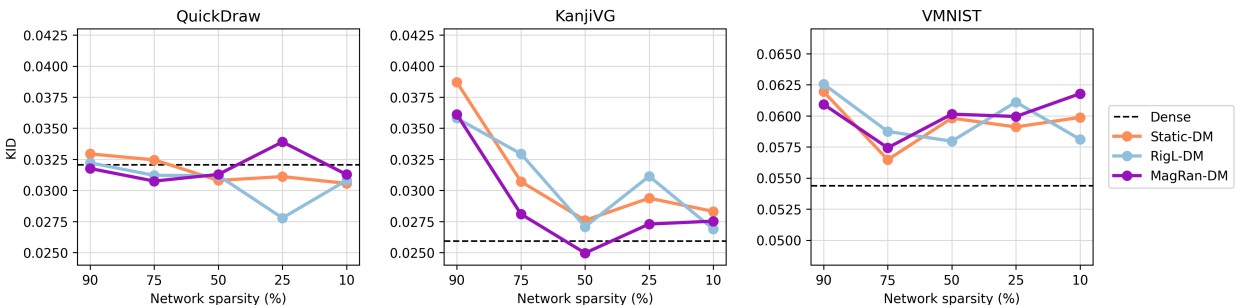

Figure 14: KID comparisons between Dense, and Static-DM, MagRan-DM and RigL-DM with various sparsity levels, for ChiroDiff, with prune and regrowth rate $p = 0.05$.

## J    Examples of Generated Samples

Figures 15 to 20 showcase examples of samples generated by Latent Diffusion and ChiroDiff across the evaluated datasets. Examples are unconditionally sampled from the Dense and the top-performing sparse model in each case. Figures 21 and 22 present examples of class-conditional generated samples on ImageNet and QuickDraw.

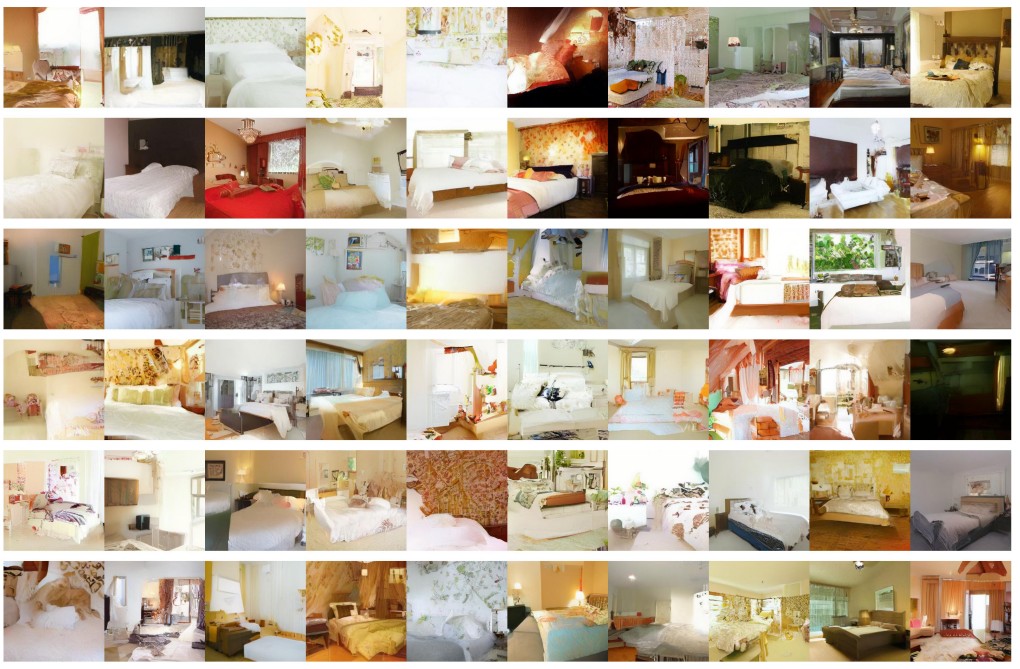

(a) Dense

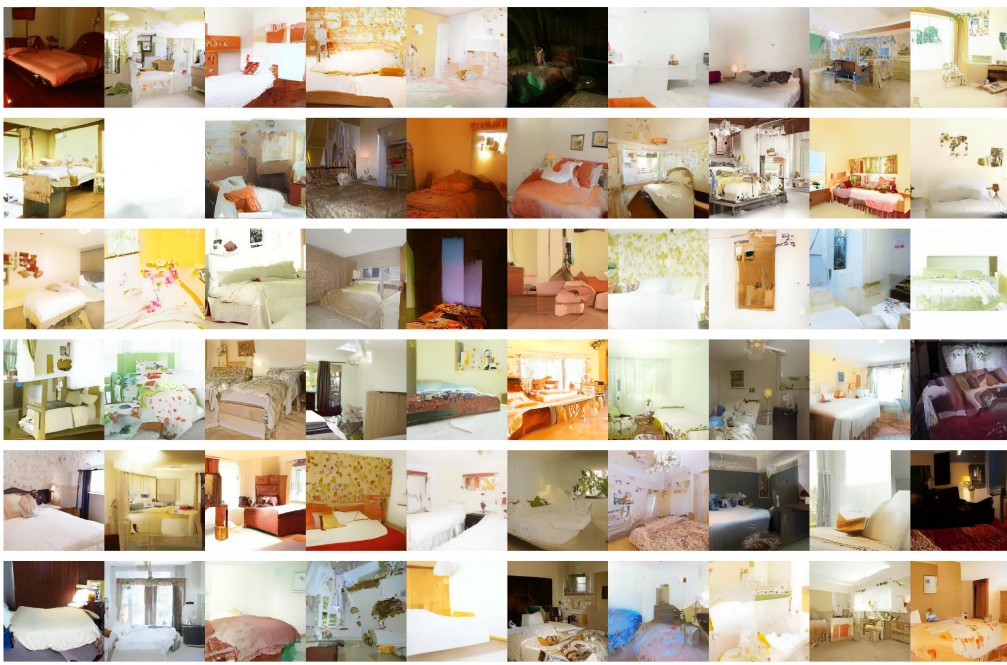

(b) Static-DM, $S = 0.25$

Figure 15: Samples from Latent Diffusion trained on LSUN-Bedrooms. The top row presents samples generated by Dense models, whereas the bottom row presents samples generated by the top-performing sparse model.

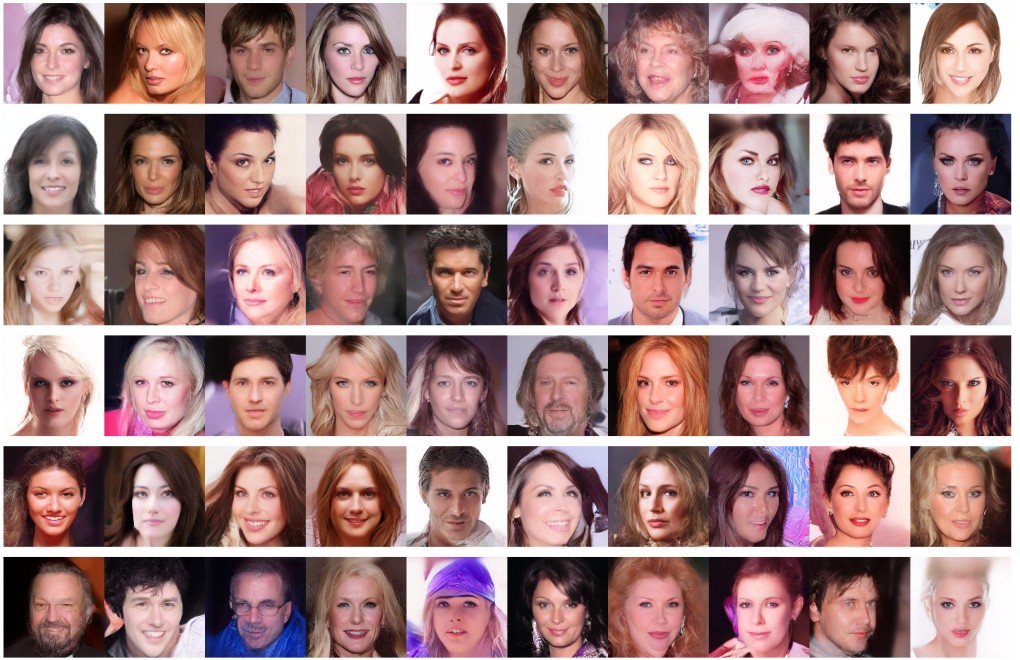

(a) Dense

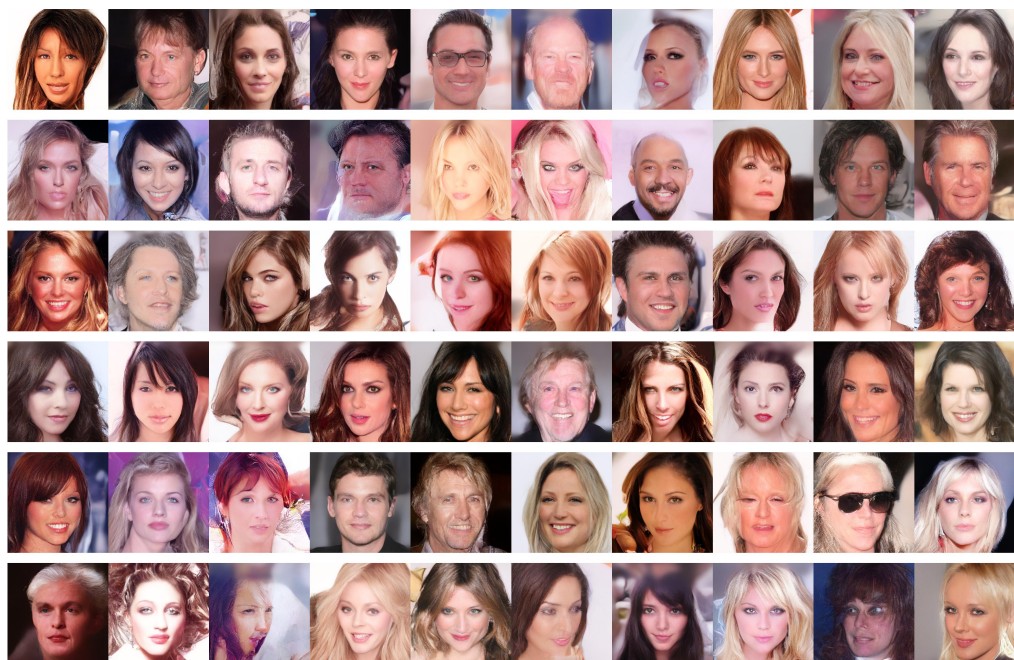

(b) RigL-DM, $S = 0.25$

Figure 16: Samples from Latent Diffusion trained on CelebA-HQ. The top row presents samples generated by Dense models, whereas the bottom row presents samples generated by the top-performing sparse model.

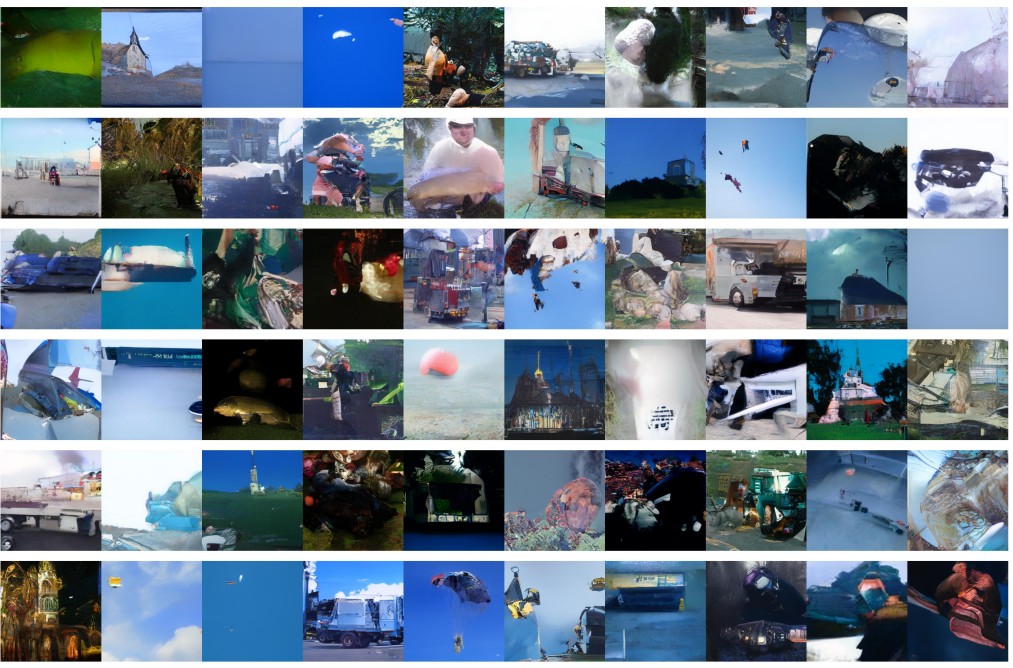

(a) Dense

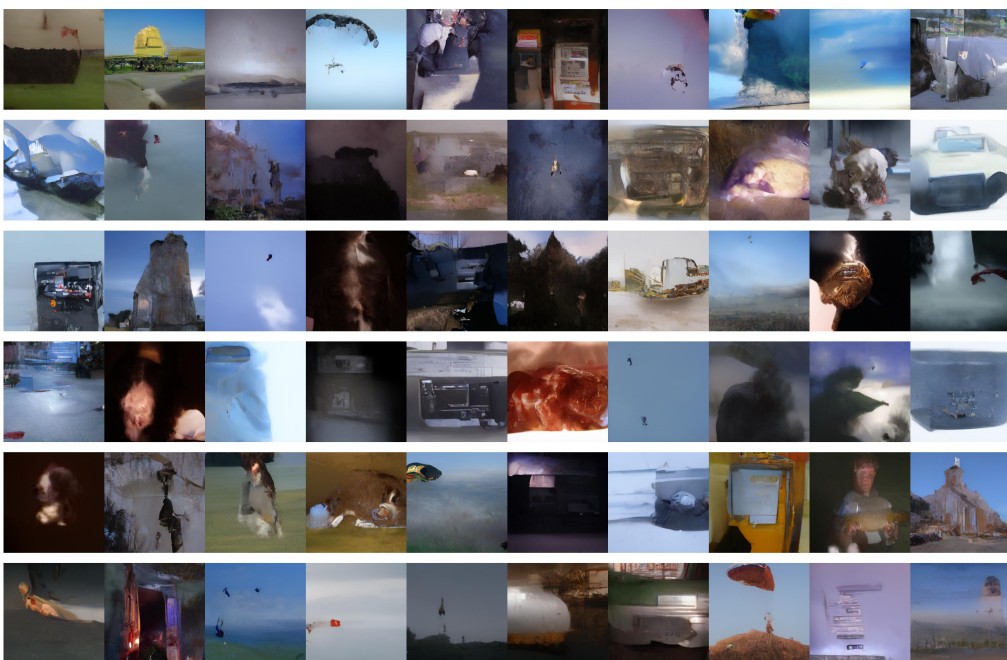

(b) MagRan-DM, $S = 0.25$

Figure 17: Samples from Latent Diffusion trained on Imagenette. The top row presents samples generated by Dense models, whereas the bottom row presents samples generated by the top-performing sparse model.

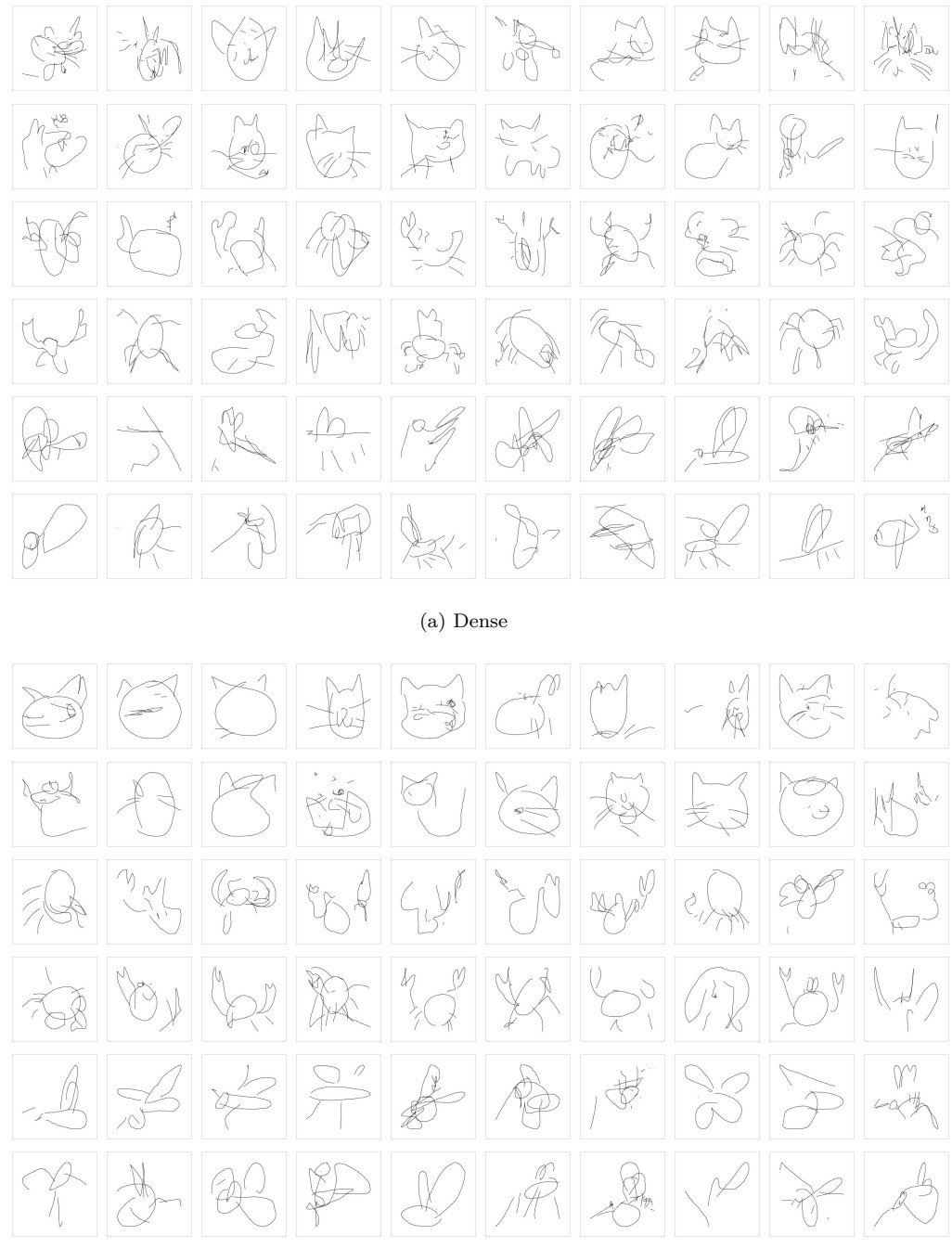

(a) Dense

(b) RigL-DM, $S = 0.10$

Figure 18: Samples from ChiroDiff trained on Quickdraw. The top row presents samples generated by Dense models, whereas the bottom row presents samples generated by the top-performing sparse model.

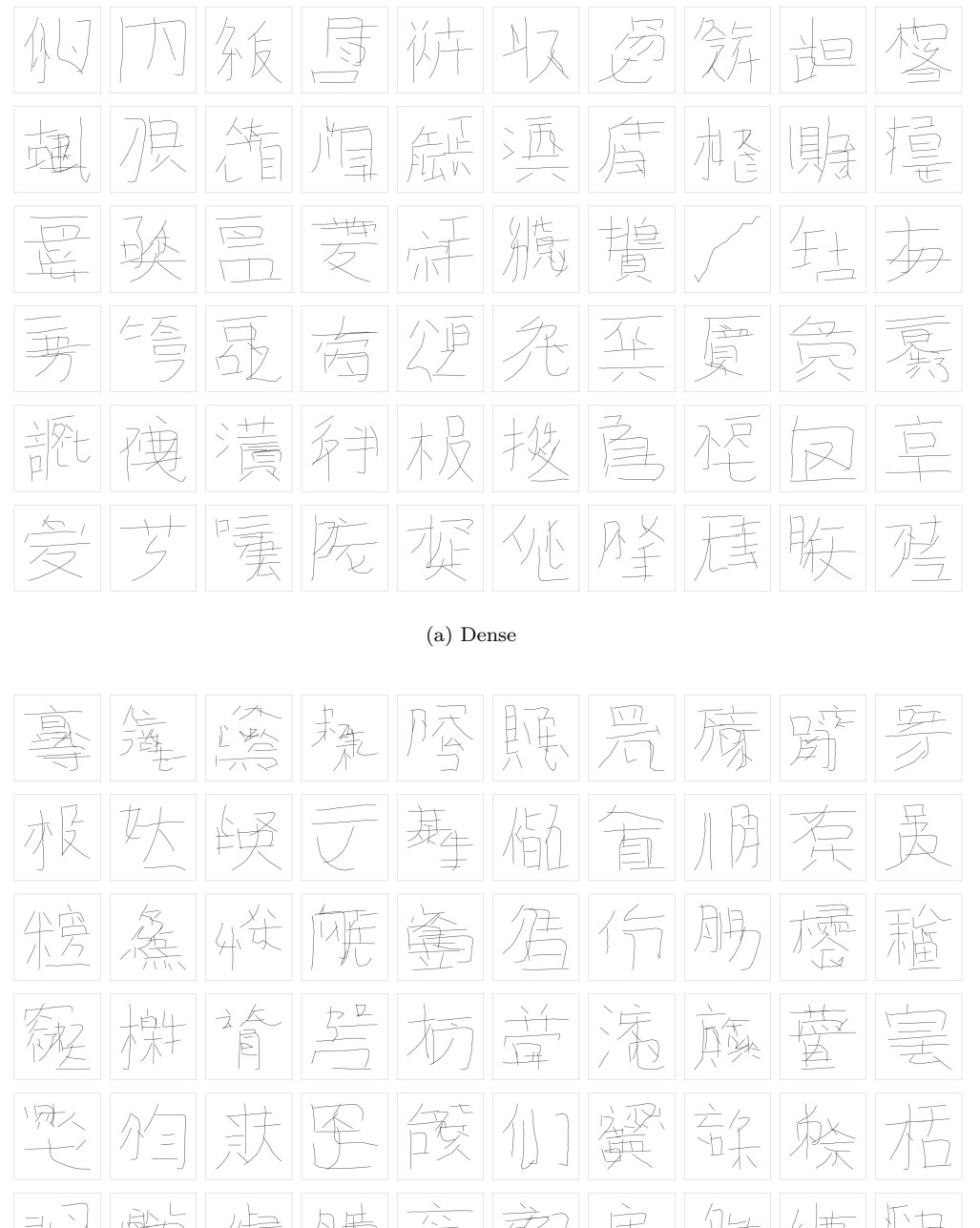

(a) Dense

(b) RigL-DM, $S = 0.25$

Figure 19: Samples from ChiroDiff trained on Kanji. The top row presents samples generated by Dense models, whereas the bottom row presents samples generated by the top-performing sparse model.

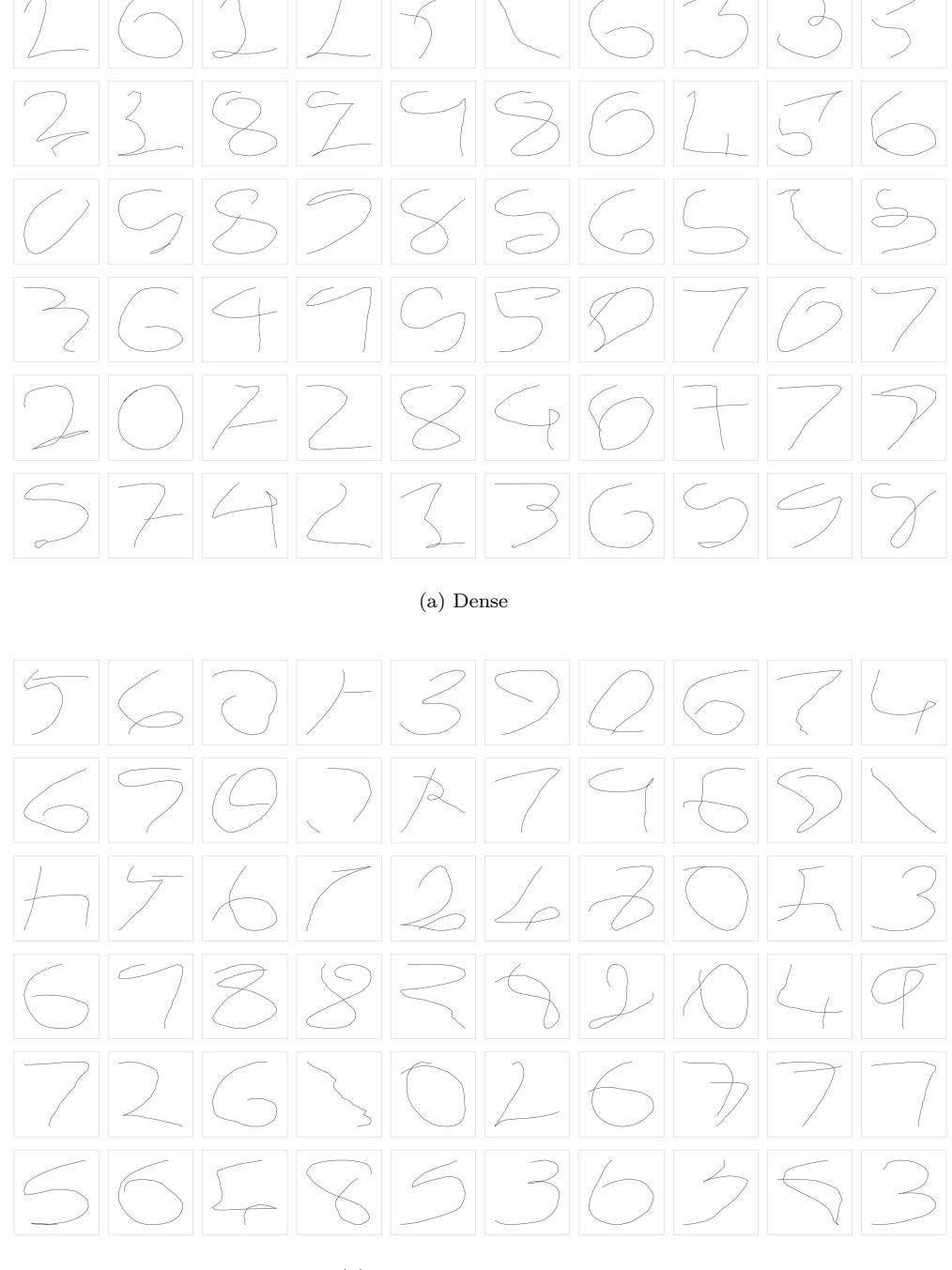

(a) Dense

(b) MagRan-DM, $S = 0.10$

Figure 20: Samples from ChiroDiff trained on VMNIST. The top row presents samples generated by Dense models, whereas the bottom row presents samples generated by the top-performing sparse model.

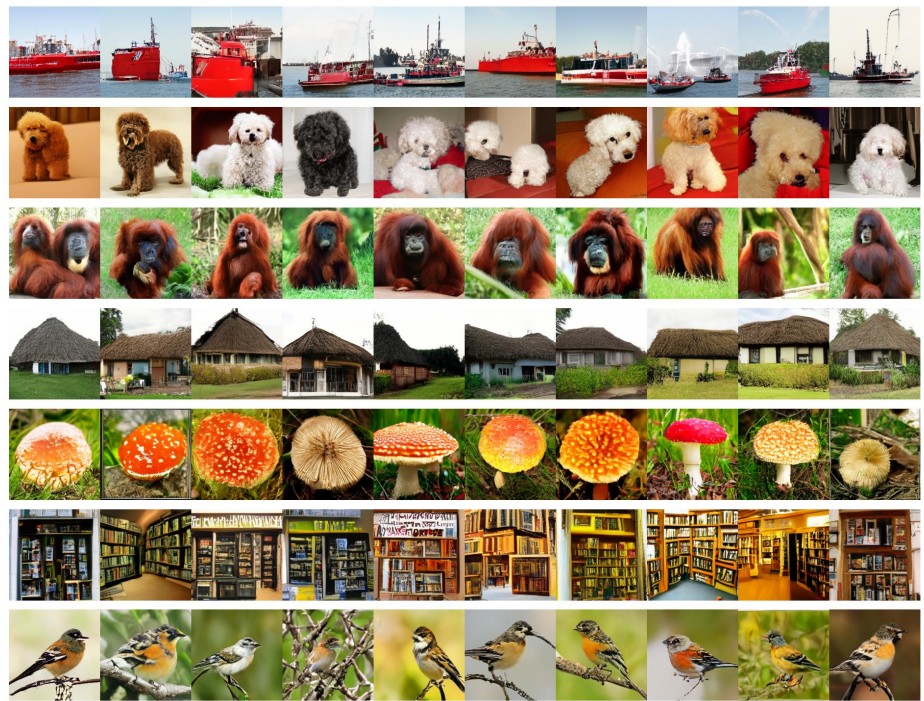

(a) Dense

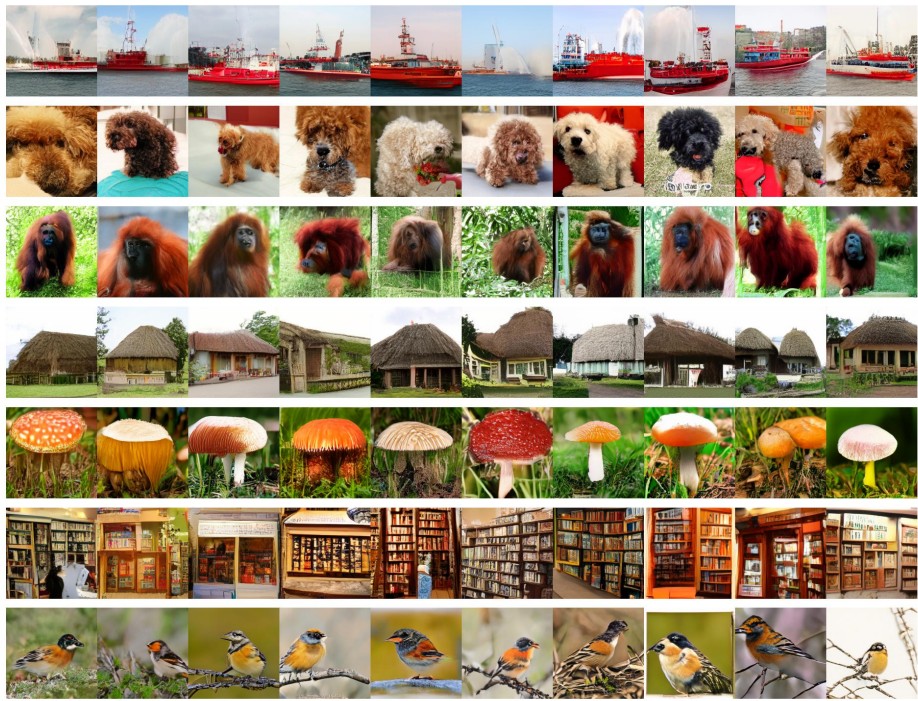

(b) MagRan-DM, $S = 0.50$

Figure 21: Samples from Latent Diffusion trained on class-conditional ImageNet. The top row presents samples generated by Dense models, whereas the bottom row presents samples generated by the top-performing sparse model.

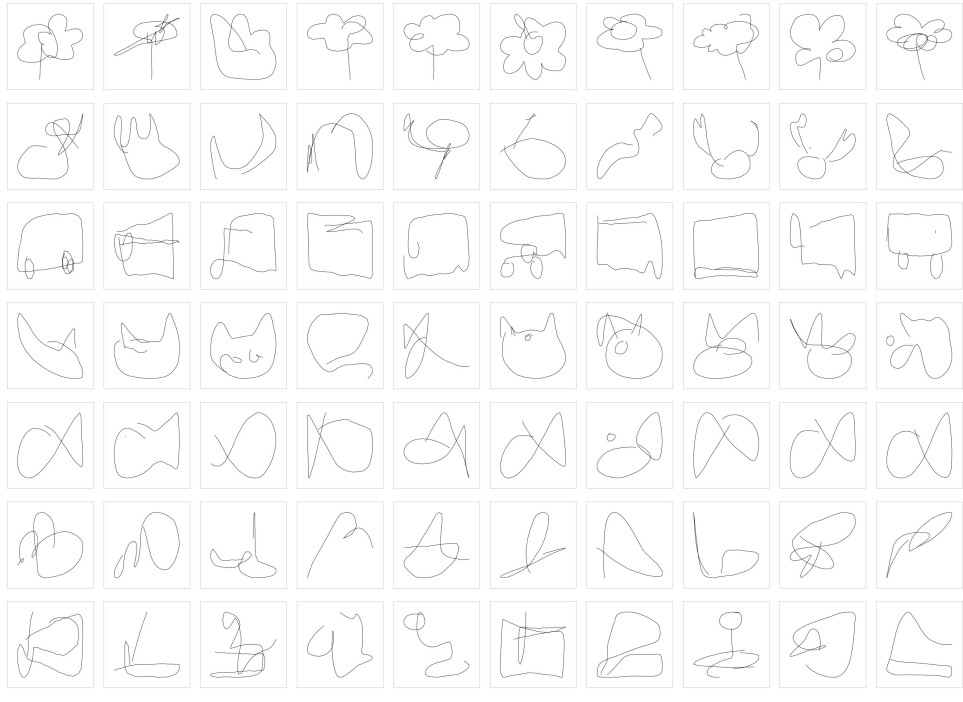

(a) Dense

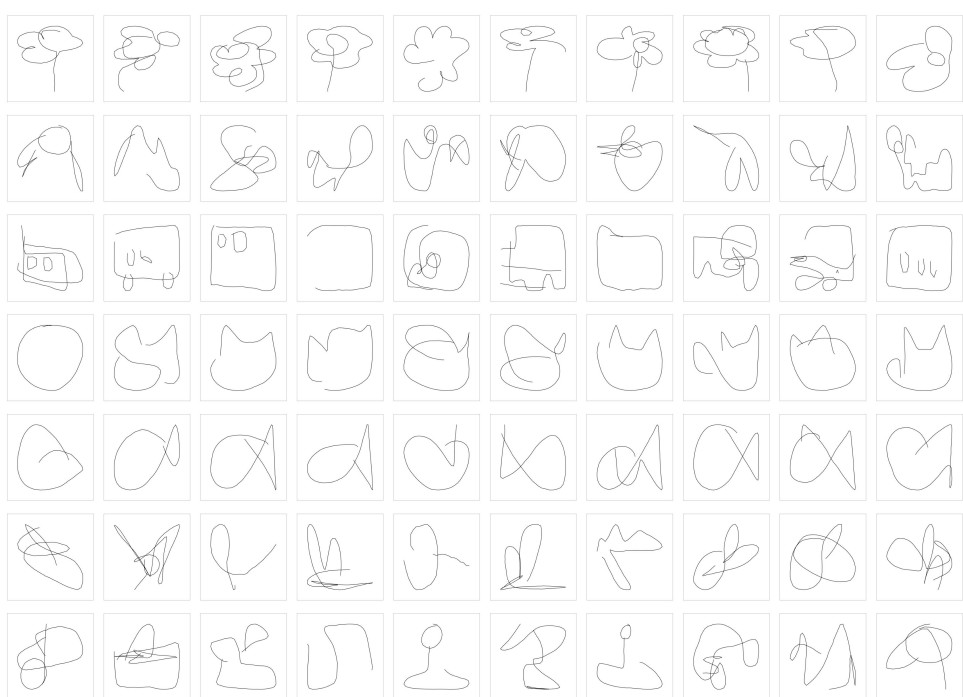

(b) MagRan-DM, $S = 0.50$

Figure 22: Samples from ChiroDiff trained on class-conditional QuickDraw. The top row presents samples generated by Dense models, whereas the bottom row presents samples generated by the top-performing sparse model.

