# OpenReview forum: "Sparse-to-Sparse Training of Diffusion Models"
_TMLR — Accepted by TMLR_

### Review · Reviewer_HyzT · 2025-06-02

**Summary Of Contributions:**

This work proposes to improve the efficiency of diffusion models in both training and inference through the sparse-to-sparse training technique. Specifically, the authors focus on the unconditional training from scratch setting and test two diffusion model variants, six datasets, and three sparse training methods. Experiments show that sparse DMs are able to match and often outperform their dense counterparts under the proposed settings.

**Audience:**

Yes

**Broader Impact Concerns:**

This work has properly discussed the broader impact.

**Claims And Evidence:**

Yes

**Requested Changes:**

Please see the weaknesses from the previous comment.

**Strengths And Weaknesses:**

S1: The idea of introducing sparse-to-sparse training into the diffusion model context is interesting and worth exploring.

S2: The paper has rather clear positioning and structures, as the goal is not to directly compare performance between models or datasets but to gain insights into the impact of sparsity in DM training.

W1: While the paper does not directly emphasize the performance between models or datasets, the current version presents rather limited analysis and insights on the impact of sparsity in DM from the perspective of dynamics. As the stable training dynamics is also one of the main advantages of DMs compared to other generative models, the impact of sparse training on learning dynamics is worth further exploration. The current main results only compare the final generation performance measured in FID, it should not be too much extra work because those models have been trained from scratch anyway.

W2: In Figure 5, the performance of the two dynamic sparse training methods, MagRan-DM and RigL-DM, shows noticeable fluctuations as the pruning ratio changes. This behavior was not evident in the main paper’s reported results. Could the authors elaborate on this discrepancy and provide further analysis?

W3: Regarding the choice of unconditional generative diffusion models, I would assume that using the classic DDPMs, rather than LDMs, would be the more intuitive and straightforward baseline—especially for a study that appears to be in the early exploratory stage. Could the authors clarify the rationale behind choosing LDMs over vanilla DDPMs?

---

> ### Author Response · Authors · 2025-07-06
> **Response to Reviewer HyzT**
>
> Dear Reviewer HyzT,
>
> Thank you for your comments and constructive review. Below we address the concerns mentioned.
>
> > **W1: While the paper does not directly emphasize the performance between models or datasets, the current version presents rather limited analysis and insights on the impact of sparsity in DM from the perspective of dynamics. As the stable training dynamics is also one of the main advantages of DMs compared to other generative models, the impact of sparse training on learning dynamics is worth further exploration. The current main results only compare the final generation performance measured in FID, it should not be too much extra work because those models have been trained from scratch anyway.**
>
> We appreciate the reviewer’s suggestion to include results on training dynamics. We have conducted experiments with the sparsity level which most consistently outperform the dense, S=0.75, and show the results in the new section 4.3 Training Dynamics. In general, sparse models appear to follow the trend of the corresponding dense model, indicating that they retain the stable training behaviour of dense diffusion models. We must note that this is only one sparsity level and one seed, due to the time constraint.
>
> >**W2: In Figure 5, the performance of the two dynamic sparse training methods, MagRan-DM and RigL-DM, shows noticeable fluctuations as the pruning ratio changes. This behavior was not evident in the main paper’s reported results. Could the authors elaborate on this discrepancy and provide further analysis?**
>
> In fact, we observed fluctuations in FID score when changing the pruning ratio. Our main results were obtained using an initial pruning ratio of 0.5. Informed by the experiments in Figure 5, we repeated the same experiments using a more conservative pruning ratio of 0.05, as mentioned in Appendix G. Our results with that pruning ratio show that a more conservative ratio improves results for high sparsity models.
>
> >**W3: Regarding the choice of unconditional generative diffusion models, I would assume that using the classic DDPMs, rather than LDMs, would be the more intuitive and straightforward baseline—especially for a study that appears to be in the early exploratory stage. Could the authors clarify the rationale behind choosing LDMs over vanilla DDPMs?**
>
> Latent Diffusion models are among the leading approaches for unconditional image generation, and are frequently used in research on diffusion models [1,2,3]. Their widespread adoption and strong efficiency led us to choose this kind of model for our study. We have clarified this in the first paragraph of Section 3.3.1.
>
> [1] Tiankai Hang, Shuyang Gu, Chen Li, Jianmin Bao, Dong Chen, Han Hu, Xin Geng, and Baining Guo. Efficient diffusion training via min-snr weighting strategy. arXiv, abs/2303.09556, 2024
>
> [2] Gongfan Fang, Xinyin Ma, and Xinchao Wang. Structural pruning for diffusion models. In Proc. Advances in Neural Information Processing Systems (NeurIPS), 2023
>
> [3] Xingyi Yang, Daquan Zhou, Jiashi Feng, and Xinchao Wang. Diffusion probabilistic model made slim. In Proc. IEEE/CVF Conference on Computer Vision and Pattern Recognition (CVPR), 2023.

---

### Review · Reviewer_fxMz · 2025-06-08

**Summary Of Contributions:**

This paper investigates sparse-to-sparse, a sparsification strategy for neural networks during training, to accelerate the training of diffusion models. The paper applies Static-DM, which prunes parameters before training, RigL-DM, which sparsifies during training, and MagRan-DM, which sparsifies while attempting reconnection during training, to two existing diffusion models, latent diffusion and ChiroDiff, and evaluates the post-training generation quality. The experiments confirm that sparse DM achieves FID scores competitive to dense DM. Additionally, the paper investigates the effectiveness of Static-DM at high sparsity levels and the impact of the number of time steps.

**Audience:**

Yes

**Broader Impact Concerns:**

Nothing to report.

**Claims And Evidence:**

No

**Requested Changes:**

- Explain the optimality of the proposed dynamic sparse training. The training dynamics of diffusion models may change significantly in the early, middle, and late training stages. DST may consistently outperform SST by adaptively changing the update rate $p$ and update frequency $\delta T$ according to the training phase (e.g., start with large changes and gradually stabilize).
- Analyze the fundamental causes of the correlation between sparsity and generation quality. Experimental results suggest that the optimal sparsity and methods vary depending on the model and dataset. For example, while performance improved with 90% sparsity in QuickDraw, it deteriorated in CelebA-HQ. The underlying mechanisms behind this are not yet fully understood. Could this performance difference be attributed to the inherent redundancy of the dataset, or to which parts of the model architecture (e.g., the attention block in U-Net, skip-connection) are more strongly affected by sparsification? Could analyzing which layers of the model are robust to sparsification and which are vulnerable lead to the design of more effective sparsification strategies?
- Please consider the possibility of sparse strategies specific to diffusion models. The sparsification method used in this study is a general-purpose method established in existing works. However, diffusion models have a unique input called noise level $t$. Would it be effective to dynamically switch sparse masks according to noise level $t$? For example, in the initial denoising steps where noise is high ($t$ is large), prioritizing dense connections to capture global structure, and in later steps where noise is low ($t$ is small), using different sparse patterns to capture details, could such time-dependent sparsification contribute to performance improvement?
- Please add experiments on conditional generation. This study does not address conditionally generated tasks such as text-to-image generation, which is important in practical applications. What impact would sparsifying mechanisms such as cross-attention, which injects conditional information into the model, have on generation quality, particularly fidelity to conditions? Could a hybrid strategy that maintains the critical pathway for transmitting conditional information while actively sparsifying other parts be effective for optimizing the efficiency of conditionally generated models?
- Please provide additional information on the characteristics of sparse-to-sparse training other than generation quality (FID). The paper focuses on the final generation quality (FID), but it does not mention the impact of sparsification on stability or convergence speed during training with a limited number of epochs. Do sparse models tend to require more time to converge or exhibit unstable learning compared to dense models? Under limited computational resources, is there a possibility of different trade-offs, such as the final performance being inferior to dense models but reaching adequate performance more quickly?
- Please clarify the differences between other previous studies, such as on GANs, and the findings of this study. When applying the sparse-to-sparse strategy, what are the differences in trends between the diffusion model and GANs?

**Strengths And Weaknesses:**

### Strengths
+ **S1.**  This paper is the first to verify efficient training using a sparse-to-sparse strategy for diffusion models.
+ **S2.** Through experiments, this paper confirms that diffusion models with sparse neural networks can achieve performance comparable to dense models.

### Weaknesses
- **W1.** As acknowledged in the paper, currently, training sparse neural networks does not benefit from hardware optimization, and even when applying the proposed method, it is not possible to reduce the training time compared to dense networks. Existing sparse-to-sparse research is based on the assumption of future hardware optimization, but the fact that no promising hardware optimization has been implemented in major frameworks even after more than five years suggests that this approach is not very promising in terms of reducing training time. To address this issue, it would be more impactful to first research hardware optimization for sparse model training rather than pursuing studies on individual models.
- **W2.** Most of the proposed methods in this paper are naive applications of existing sparse-to-sparse strategies and lack technical novelty. In particular, dynamic sparsification (RigL-DM and MagRan-DM) is applied naively without considering the time step $t$, making it almost obvious that these methods are inferior to Static-DM because conditioning different $t$ makes models activate different neurons.
- **W3.** It is already known from previous studies, such as GAN, that sparse-to-sparse can achieve high performance while reducing FLOPS, but the experimental results shown in this paper lack insights specific to diffusion models.
- **W4.** There is little evidence to support the claim that sparse-to-sparse training in diffusion models is essential for achieving the optimal trade-off between performance and parameter reduction. For example, it is unclear whether the proposed method is more effective than dense models with the same parameter size after pruning.

---

> ### Author Response · Authors · 2025-07-06
> **Response to Reviewer fxMz**
>
> Dear Reviewer fxMz,
>
> Thank you for your insightful comments and feedback. We have addressed your concerns as outlined below.
>
> >**1. Explain the optimality of the proposed dynamic sparse training. The training dynamics of diffusion models may change significantly in the early, middle, and late training stages. DST may consistently outperform SST by adaptively changing the update rate and update frequency according to the training phase (e.g., start with large changes and gradually stabilize).**
>
> As stated in the Introduction, to the best of our knowledge, our work is the first to explore this problem of sparse-to-sparse training for Diffusion Models. Therefore, our objective in this work is to establish a first baseline for how sparsity behaves within diffusion models, using standard techniques with minimal tuning.  The suggestion of adapting sparse training hyperparameters during training could be an interesting direction of future research and as such we have indicated it in Section 4.5.
>
> >**2. Analyze the fundamental causes of the correlation between sparsity and generation quality. Experimental results suggest that the optimal sparsity and methods vary depending on the model and dataset. For example, while performance improved with 90% sparsity in QuickDraw, it deteriorated in CelebA-HQ. The underlying mechanisms behind this are not yet fully understood. Could this performance difference be attributed to the inherent redundancy of the dataset, or to which parts of the model architecture (e.g., the attention block in U-Net, skip-connection) are more strongly affected by sparsification? Could analyzing which layers of the model are robust to sparsification and which are vulnerable lead to the design of more effective sparsification strategies?**
>
> We appreciate the reviewer’s insightful questions. Indeed, this difference in performance might be related to the factors suggested. There are many mechanisms in sparse-to-sparse training that are not yet fully understood, even in the more extensively studied setting of supervised learning, where many papers exist on this topic. These complexities merit a comprehensive and systematic investigation. In the scope of this paper, a focused investigation on the impact of sparsity on specific layers can be performed and would provide valuable insights. Given additional time, we would be happy to include these experiments.
>
> >**3. Please consider the possibility of sparse strategies specific to diffusion models. The sparsification method used in this study is a general-purpose method established in existing works. However, diffusion models have a unique input called noise level . Would it be effective to dynamically switch sparse masks according to noise level ? For example, in the initial denoising steps where noise is high ( is large), prioritizing dense connections to capture global structure, and in later steps where noise is low ( is small), using different sparse patterns to capture details, could such time-dependent sparsification contribute to performance improvement.**
>
> We thank the reviewer for this interesting suggestion. Dynamically adapting sparse masks according to the noise levels is a very promising idea, which has been explored by [1], albeit not by using sparse-to-sparse training. We will mention this in Section 4.4.
>
> [1] Lexington Whalen, Zhenbang Du, Haoran You, Chaojian Li, Sixu Li, and Yingyan (Celine) Lin. Early-Bird Diffusion: Investigating and Leveraging Timestep-Aware Early-Bird Tickets in Diffusion Models for Efficient Training. In Proc. The IEEE/CVF Conference on Computer Vision and Pattern Recognition, 2025
>
> >**4. Please add experiments on conditional generation. This study does not address conditionally generated tasks such as text-to-image generation, which is important in practical applications. What impact would sparsifying mechanisms such as cross-attention, which injects conditional information into the model, have on generation quality, particularly fidelity to conditions? Could a hybrid strategy that maintains the critical pathway for transmitting conditional information while actively sparsifying other parts be effective for optimizing the efficiency of conditionally generated models?**
>
> We thank the reviewer for this suggestion, also suggested by Reviewer JRjh. We agree that demonstrating conditional image generation would strengthen our paper. We are still running these experiments and will request an extension (beyond the default 2 weeks) in order to report them in the paper.

---

> ### Author Response · Authors · 2025-07-07
> **Continuation of Response to Reviewer fxMz**
>
> >**5. Please provide additional information on the characteristics of sparse-to-sparse training other than generation quality (FID). The paper focuses on the final generation quality (FID), but it does not mention the impact of sparsification on stability or convergence speed during training with a limited number of epochs. Do sparse models tend to require more time to converge or exhibit unstable learning compared to dense models? Under limited computational resources, is there a possibility of different trade-offs, such as the final performance being inferior to dense models but reaching adequate performance more quickly?**
>
> We appreciate and agree with the reviewer’s suggestion. Regarding training dynamics, we have conducted experiments with the sparsity level which most consistently outperform the dense, S=0.75. The results can be seen in the new section 4.3 Training Dynamics. In general, sparse models appear to follow the trend of the corresponding dense model, indicating that they retain the stable training behaviour of dense diffusion models. We must note that this is only one sparsity level and one run (one random seed), due to the time constraint of 2 weeks. We will report 3 runs in the final version of the paper, for consistency with the rest of our experiments.
>
> >**6. Please clarify the differences between other previous studies, such as on GANs, and the findings of this study. When applying the sparse-to-sparse strategy, what are the differences in trends between the diffusion model and GANs?**
>
> >**W3. It is already known from previous studies, such as GAN, that sparse-to-sparse can achieve high performance while reducing FLOPS, but the experimental results shown in this paper lack insights specific to diffusion models.**
>
> While sparse-to-sparse training has been explored previously in the generative modelling paradigm in the context of GANs, our work focuses specifically on Diffusion Models, which are completely different models. A detailed comparison of trends between GANs and diffusion models would warrant a new research paper.
>
>
> >**W1. As acknowledged in the paper, currently, training sparse neural networks does not benefit from hardware optimization, and even when applying the proposed method, it is not possible to reduce the training time compared to dense networks. Existing sparse-to-sparse research is based on the assumption of future hardware optimization, but the fact that no promising hardware optimization has been implemented in major frameworks even after more than five years suggests that this approach is not very promising in terms of reducing training time. To address this issue, it would be more impactful to first research hardware optimization for sparse model training rather than pursuing studies on individual models.**
>
> We respectfully disagree. While it is true that, to date, NVIDIA GPUs only support fine-grained structured sparsity, companies such as Cerebras have made significant progress in developing hardware that supports unstructured sparsity, both dynamic and static. We believe that research on hardware optimization and the effect of sparsity in specific models can proceed in parallel, and drive progress together towards more efficient neural network training.
>
> >**W2. Most of the proposed methods in this paper are naive applications of existing sparse-to-sparse strategies and lack technical novelty. In particular, dynamic sparsification (RigL-DM and MagRan-DM) is applied naively without considering the time step t, making it almost obvious that these methods are inferior to Static-DM because conditioning different  t makes models activate different neurons.**
>
> We appreciate the reviewer’s observations. Our work is the first to verify that it is indeed possible to use sparse-to-sparse training for diffusion models, using standard techniques with minimal tuning.
>
> We acknowledge that we do not explicitly consider the timestep t during the weight exploration process for DST techniques. However, since each batch contains multiple timesteps, and pruning and regrowth are performed based on this collection of timesteps, we do not see how this would lead to significant issues. In fact, our results do not show a clear superiority of Static-DM over DST techniques. We would appreciate it if the reviewer could kindly clarify their concern regarding this point.
>
> >**W4. There is little evidence to support the claim that sparse-to-sparse training in diffusion models is essential for achieving the optimal trade-off between performance and parameter reduction. For example, it is unclear whether the proposed method is more effective than dense models with the same parameter size after pruning.**
>
> We would like to clarify that the primary focus of our work is to improve training efficiency. Pruning a dense model to reduce its parameters still requires training a fully dense version. In contrast, our methods aim to reduce DMs’ high computational cost incurred during training.

---

### Review · Reviewer_JRjh · 2025-06-22

**Summary Of Contributions:**

This paper introduces the concept of sparse-to-sparse training for diffusion models, enabling models to be trained from scratch with fixed or sparse connectivitons that evolves dynamically. It proposes three methods: Static-DM, MagRan-DM, and RigL-DM and evaluates them on two diffusion architectures across multiple datasets. The study demonstrates that sparse models can match or exceed dense performance while significantly reducing parameters and FLOPs. This work provides insights into efficient training strategies for generative models, addressing both computational cost and scalability.

**Audience:**

Yes

**Broader Impact Concerns:**

There is a broader impact statement provided by the authors which satisfactorily addresses the concerns of misuse of generative content.

**Claims And Evidence:**

Yes

**Requested Changes:**

1. It would be interesting if the authors include runtime benchmarks on sparse-compatible hardware like A100 to empirically justify the practical efficiency of the method.
2. It would be great if the authors provide one experimental result on conditional image generation. Perhaps a text-to-image task using LDM
3. Ablation studies on initialization strategies is appreciated.
4. The outperformance of the sparse model (magran-dm at 90% sparsity) over dense model raises a red flag. Perhaps scale down the dense model's width or depth so that it has a comparable number of params or FLOPs as the sparse model for fairness.
5. Results on other evaluation metrics will be appreciated.

**Strengths And Weaknesses:**

Strengths:
1. The paper is concise, well-written and easy to understand
2. The work demonstrates substantial reduction in FLOPs and parameters.
3. The experiments are conducted over a wide variety of datasets and span multiple sparsity methods.

Weakenesses:
1. No runtime benchmarks on sparse-compatible hardware, although there is discussion about A100 hardware. So, the efficiency gains remain theoretical.
2. Conditional generation tasks are not explored. It would be interesting to see how the sparse training affects given a prompt of any modality.
3. The method heavily relies on initialization and DST hyperparameters like prune/growth rate. A bit of ablation of these components would be interesting to see.
4. In the chirodiff experiment on quickdraw dataset, the extremely sparse model also outperforms the dense ones. Is it possible that the dense model is not calibrated properly?
5. The evaluation is only reported in terms of FID. It would be interesting to see other evaluation metrics like Inception Score or similar generative metrics.

---

> ### Author Response · Authors · 2025-07-06
> **Response to Reviewer JRjh**
>
> Dear reviewer JRjh
>
> Thank you for your constructive feedback. Please find our responses to the concerns mentioned below.
>
> >**1. It would be interesting if the authors include runtime benchmarks on sparse-compatible hardware like A100 to empirically justify the practical efficiency of the method.**
>
> We appreciate the reviewer’s suggestion. However, A100’s hardware acceleration supports only  fine-grained structured sparsity in a 2:4 pattern, while our methods rely on unstructured sparsity, which A100 cannot handle. Other hardware such as the Cerebras CS-2 and CS-3 do support the acceleration of models with unstructured sparsity, both dynamic and static. Unfortunately, we do not have access to this hardware. We have reached out to Cerebras but have received no reply yet.
>
> >**2. It would be great if the authors provide one experimental result on conditional image generation. Perhaps a text-to-image task using LDM.**
>
> We thank the reviewer for this suggestion, and agree that demonstrating conditional image generation would strengthen our paper. We are still running these experiments and will request an extension (beyond the default 2 weeks) in order to report them in the paper.
>
> >**3. Ablation studies on initialization strategies is appreciated.**
>
> We would kindly ask for clarification on this request. There are many possible ways of initializing sparse models, and we follow the most commonly used approach in the literature [1, 2, 3].
>
> [1] Shiwei Liu, Yuesong Tian, Tianlong Chen, and Li Shen. Don’t be so dense: Sparse-to-sparse GAN training without sacrificing performance. International Journal of Computer Vision, 2023b.
>
> [2] Utku Evci, Trevor Gale, Jacob Menick, Pablo Samuel Castro, and Erich Elsen. Rigging the lottery: Making all tickets winners. In Proc. International Conference on Machine Learning (ICML), 2020.
>
> [3] Shiwei Liu, Lu Yin, Decebal Constantin Mocanu, and Mykola Pechenizkiy. Do we actually need dense over-parameterization? In-time over-parameterization in sparse training. arXiv, abs/2102.02887, 2021c.
>
>
> >**4. The outperformance of the sparse model (magran-dm at 90% sparsity) over dense model raises a red flag. Perhaps scale down the dense model's width or depth so that it has a comparable number of params or FLOPs as the sparse model for fairness.**
>
> We understand the concern, but extremely sparse models matching or outperforming the performance of dense models is a behaviour that has been observed before in other types of models, such as MLPs and CNNs [1, 2]. Please note that our goal was not to tune the baseline models but to compare dense and sparse versions of each model, all things being equal. However, we cannot rule out the possibility that the original dense model is not calibrated properly, so we have conducted experiments for the QuickDraw dataset using a smaller version of the model. These results can be observed in Appendix E. Overall, with a smaller model, sparse versions show less comparable performance to their dense counterparts at higher sparsity levels, exhibiting the same downward trend of diminishing performance with increased sparsity as the other datasets.
>
> [1] Decebal Mocanu, Elena Mocanu, Peter Stone, Phuong Nguyen, Madeleine Gibescu, and Antonio Liotta. Scalable training of artificial neural networks with adaptive sparse connectivity inspired by network science. Nature Communications, 2018.
>
> [2] Utku Evci, Trevor Gale, Jacob Menick, Pablo Samuel Castro, and Erich Elsen. Rigging the lottery: Making all tickets winners. In Proc. International Conference on Machine Learning (ICML), 2020.
>
> >**5. Results on other evaluation metrics will be appreciated.**
>
> We appreciate and agree with the reviewer’s suggestion. In addition to FID, we report Kernel Inception Distance (KID) for a more comprehensive evaluation. These results can be seen in the new Appendix H.

---

### Comment · Reviewer_JRjh · 2025-07-29
**Official Comment to the Response by the Authors**

1. Thank you for the clarification.
2. I am inclined to provide an extension given that the authors provide the results that I requested.
3. In the weakness section I mentioned that "The method heavily relies on initialization and DST hyperparameters like prune/growth rate. A bit of ablation of these components would be interesting to see." I would like to see how the model performance varies by varying the DST hyperparams.
4. Thank you for the clarification.
5. Appreciate the added result, is it possible to add Inception Score result as well?

---

### Decision · Action_Editor_Cbdn · 2025-08-24

**Recommendation:** Accept as is

**Audience:**

Yes

**Audience Explanation:**

Computational efficiency (both training and inference) of diffusion models continues to remain a critical aspect and the paper makes a timely contribution on this topic.

**Claims And Evidence:**

Yes

**Claims Explanation:**

This paper introduces sparse-to-sparse training for diffusion models with the goal of addressing computationally expensive training and inference in these models.

Two of the three reviewers were positive about the paper and emphasized that the paper is well-motivated and well-executed on various aspects including novelty and empirical rigor, and is likely to stimulate further research on the topic. One of these reviewers also acknowledged the additional analysis of training dynamics adequately which addressed their initial concerns, and stressed that the sparse training perspective offers a valuable alternative to distillation-based efficiency methods.

One of the reviewers expressed concerns about perceived limits in technical novelty and incomplete experiments (e.g., conditional generation). The reviewer also remarked that prior works on sparse architectures have not led to hardware accelerations in practice. The authors responded to these concerns in detail.

The novelty aspect is not a deal-breaker for TMLR as the paper does seem to deliver in terms of the claimed contributions, as the other reviewers also agree. The experimental results demonstrate consistent efficiency gains and competitive performance, validating the method’s importance even if some extensions (like conditional generation) remain for future work.

The paper addresses a pressing challenge in generative modeling and offers a new research direction that complements other approaches such as distillation. Although it is not critical, I advise (as some other reviewers also suggested) the authors to include an additional experiment on conditional generation in the final version. That would strengthen the paper further.